

# CO₂ and hydrography acquired by Autonomous Surface Vehicles from the Atlantic Ocean to the Mediterranean Sea: data correction and validation

Riccardo Martellucci[1], Michele Giani[1], Elena Mauri[1], Laurent Coppola[2], Melf Paulsen[3], Marine Fourrier[2], Sara Pensieri[4], Vanessa Cardin[1], Carlotta Dentico[5], Roberto Bozzano[4], Carolina Cantoni[6], Anna Luchetta[6], and Ingunn Skjelvan[7]

[1]National Institute of Oceanography and Applied Geophysics, Trieste, Italy

[2]Oceanography Laboratory of Villefranche, Villefranche, France

[3]GEOMAR Helmholtz Centre for Ocean Research Kiel, Kiel, Germany

[4]National Research Council - Institute for the study of Anthropic Impact and Sustainability in the Marine Environment (CNR-IAS), Genova, Italy

[5]Department of Environmental Sciences, Informatics and Statistics, Università Cà Foscari, Venice,, Italy

[6]National Research Council-Institute of Marine Sciences (CNR-ISMAR), Trieste, Italy

[7]NORCE Norwegian Research Centre, Bjerknes Centre for CLimate Research. Beregn, Norway

Corresponding author: Riccardo Martellucci (rmartellucci@ogs.it)

**Abstract**

The ATL2MED demonstration experiment involved two autonomous surface vehicles provided by Saildrone Inc. (SD) along a route from the tropical eastern North Atlantic to the Adriatic Sea between October 2019 and July 2020. This nine-month long experiment located in a transition zone between the temperate and tropical belts represents a major challenge in the use of SD. The sensors on board were subjected, to varying degradation degrees depending on the geographical area and the season, to biofouling with consequent deterioration of the acquired measurements. As a result, several maintenance along the mission's course were necessary.

We address the difficulty of correcting the data during a period of COVID-19 restrictions, which significantly reduced the number of discrete samples planned for SD salinity and dissolved oxygen validation. This article details alternative correction methods for salinity and dissolved oxygen. Due to the lack of in situ data, model products have been used to correct the salinity data acquired by the SDs, and then the resulting corrected salinity was validated with data from fixed ocean stations, gliders, and Argo floats. In addition, dissolved oxygen data acquired from SDs after correction using air oxygen measurements were tested and found to be in line with the oxygen values expected from temperature and chlorophyll-a data. The correction methods are relevant and useful in situations where validation capabilities are lacking, which was the case during the ATL2MED demonstration experiment. For future experiments, it is recommended that validation samples are collected more frequently. An overview over data availability is found in Section 5, Table 8.

## 1    Introduction

Automated observations contribute to a steadily increasing knowledge of the ocean and its role in the global climate system. For a long time, fixed ocean stations and research vessels formed the backbone of the monitoring network.



In recent years, efforts have been made to improve the frequency of acquisition through technological developments (e.g., EU infrastructures ICOS, https://www.icos-cp.eu/; EMSO, https://emso.eu; EuroArgo, https://www.euro-argo.eu) where fixed ocean stations and ships of opportunity are equipped with autonomous and accurate partial pressure of $CO_2$ ($pCO_2$) sensors in addition to sensors for complementary measurements (e.g., water temperature, salinity, dissolved oxygen, pH, nutrients, fluorescence) that are necessary to understand the dynamics and the effects of $CO_2$ fluxes on the carbon budget. Despite the relevant efforts, it is still difficult to obtain a comprehensive overview of $CO_2$ flux at regional and larger scale, because of very sparse coverage of fixed observatories, low measurement frequency, and limited systematic reference measurements.

One way to address this gap is to develop and deploy Autonomous Surface Vehicles (ASVs) equipped with a suite of sensors, and capable of measuring $CO_2$ fluxes at the air-sea interface with gas reference, high sampling frequency and real-time data transmission. ASV monitoring systems have the potential to collect data from large ocean areas and at a frequency that resolves processes at multiple time scales. Nevertheless, there are challenges with those surface monitoring systems, and one of the most important is the biofouling, which can interfere with measurements of e.g., conductivity, dissolved oxygen and especially chlorophyll-a (Chl-a), and could ultimately render the sensors inoperable. Regular maintenance counteracts biofouling or at least reduces the impact on measurements, but this is not always possible due to long distance from shore or from the maintenance vessel. Therefore, the value of ASV data depends heavily on quality control and quality assurance.

During the 9 month long demonstration experiment ATL2MED, two ASVs provided by Saildrone Inc. (SD) were used to improve data coverage and link $CO_2$ surface observations at fixed ocean stations on a larger scale from the eastern tropical North Atlantic to the central Mediterranean Sea. SDs are prone to errors primarily due to sensor drift, which can be caused by either biofouling or problems with the sensor itself. During the ATL2MED demonstration experiment, problems were found with the data collected by several SD sensors, and severe biofouling occurred, as expected in such a long duration experiment.

Still, the use of SDs provided the opportunity to expand and link fixed $CO_2$ observations at the surface on a larger scale, particularly during the COVID-19 pandemic when access to ocean platforms and ship visits were restricted or even prohibited. Furthermore, the demonstration experiment allowed us to focus SD measurements on different marine ecosystems using multiple seasons, which made it possible to assess the quality of measurements across a wide range of values. The experiment additionally evaluated the ability of such ASVs to provide high-quality data for the scientific community.

The objective of the present work is to evaluate and correct the data collected by the SDs and to provide a homogenised and comparable dataset that is useful to study processes in the Atlantic Ocean and Mediterranean Sea.

## 2 Experiment and data

The ATL2MED demonstration experiment took place between October 2019 and July 2020 as a joint effort among a number of European academic institutions (GEOMAR Helmholtz Centre for Ocean Research (GEOMAR), the French National Centre for Scientific Research (CNRS), Oceanography Laboratory of Villefranche (LOV), the Oceanic Platform of the Canary Islands (PLOCAN), Ocean Science Centre Mindelo (OSCM), the Hydrographic Institute of Portugal (IH), Balearic Islands Coastal Observing and Forecasting System (SOCIB), Italian National Institute of Oceanography and Applied Geophysics (OGS), Helmholtz Zentrum Geesthacht (HZG), Centre Scientifique de Monaco (CSM), National Research Council-Institute of Marine Sciences (CNR-ISMAR), National Research Council- Institute for the study of Anthropic Impact and Sustainability in the Marine Environment (CNR-IAS), the European infrastructure Integrated Carbon Observation System - Ocean Thematic Centre (ICOS-OTC), and the U.S. company Saildrone Inc. During the experiment, the SDs crossed ocean areas with specific characteristics. The track covered the eastern tropical North Atlantic, the Strait of Gibraltar, and the northern part of the western and central Mediterranean Sea including the Ligurian Sea, the Strait of Sicily, the Strait of Otranto, and the Adriatic Sea (Fig. 1).

The demonstration experiment included not only sensors and instruments installed at the SDs, but also equipment deployed at a number of facilities. Data collected by these facilities were used to correct data from the SDs (see Section 3), and the different facilities and instruments are presented below. We focus on data collected

at 5 fixed ocean stations (DYFAMED, W1M3A, E2M3A, PALOMA, and MIRAMARE), 2 gliders (Nice-Calvi
section, southern Adriatic section), and one research vessel (R/V Meteor). Table 1 gives an overview of the
different facilities and when they were visited by the SDs, while Table 2 indicate when SD maintenance were
performed. Tables 3 and 4 list the different instruments and sensors, their location, main characteristics, and
frequency of measurements. Detailed description about the ATL2MED demonstration experiment is available in
Skjelvan et al. (2021).

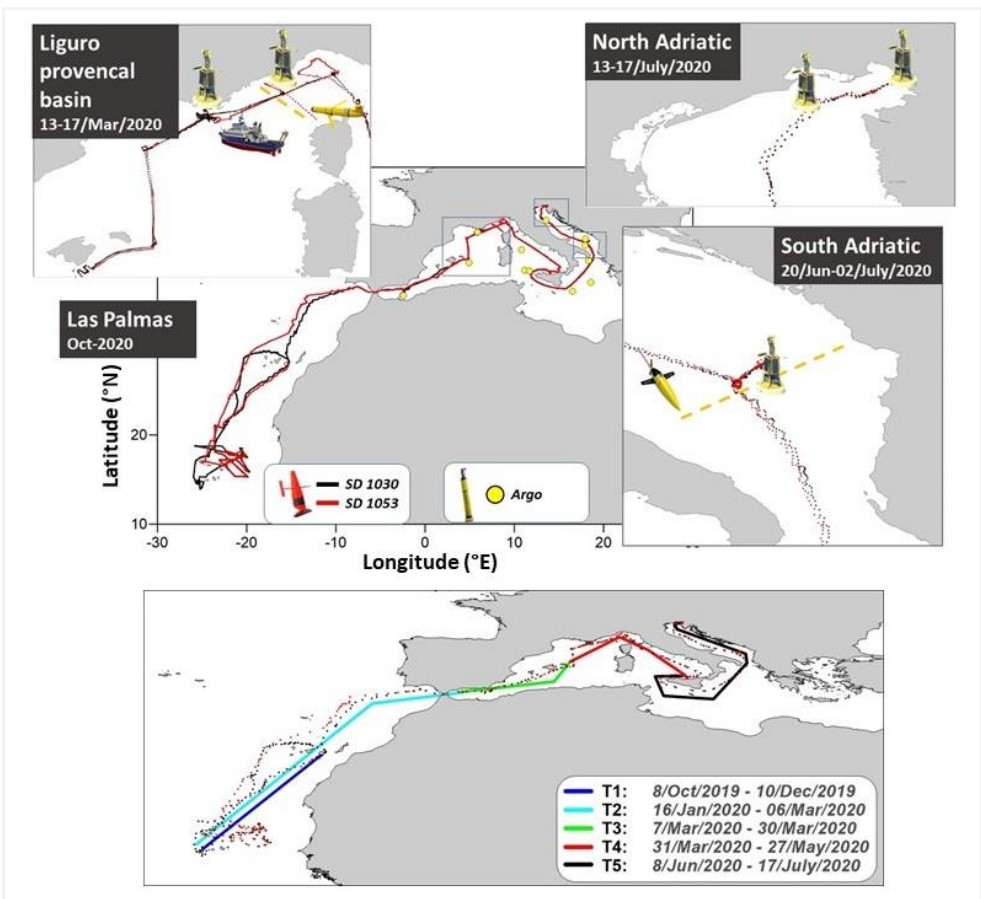

**Figure 1. Upper maps show the sailing route of the two SDs (black and red lines), positions of Argo floats (yellow dots), and positions of the fixed stations (buoy sketches). The lower map shows the SD routes divided into transects (T) based on the maintenances that were performed during the experiment.**


**2.1    Saildrones (SD)**
The SDs, provided by Saildrone Inc. were equipped with a number of autonomous sensors (CTD, dissolved
oxygen, fluorescence, pH, $pCO_2$ and meteorological sensors) and here we focus primarily on the sensors that
regularly measured temperature, salinity, dissolved oxygen, and $pCO_2$. This selection is based on available options
for correcting the SD datasets: some of the sensors (e.g., fluorescence) were affected by biofouling to such a degree
that it could not be accounted for and others worked only for a short period of time (e.g., Durafet Honeywell pH
sensor). One of the SDs (SD 1030) was equipped with an ASVCO2 system developed by PMEL (NOAA's Pacific
Marine Environmental Laboratory). The ASVCO2 system is a compressed version of the more voluminous system



described in detail in Sutton et al. (2014). Water from approximately 0.5 m depth enters a bubble equilibrator
(Friederich et al., 1995) and partially dried $xCO_2$ is measured with an infrared detector (LI-COR 820 CO2 gas
analyser). Two points calibration was used where the first is a reference gas from NOAA/ERSL while the second
is air purged for $CO_2$. An air inlet was mounted approximately 1 m above sea level and atmospheric $xCO_2$ was
measured in between the sea surface measurements. The measurement frequency during the ATL2MED
experiment was once per hour.

**2.2      LION and DYFAMED fixed stations**
In the French EEZ, the open fixed stations LION and DYFAMED are located in the Gulf of Lion and in the
Ligurian Sea in the northwestern Mediterranean Sea, respectively. CNRS and LOV are in charge of the stations,
which are equipped with ODAS surface buoys (Météo France) capable of measuring surface ocean hydrography
(temperature and salinity) and meteorology. At the DYFAMED site, these observations are complemented by a
second surface buoy (BOUSSOLE) for autonomous $pCO_2$ measurements, monthly ship visits for in situ
measurements (CTD profile and Niskin samples) and regular deployment of ocean gliders performing the Nice-
Calvi section from January to the end of May (MOOSE program; Coppola et al., 2019). The $pCO_2$ measurements
at DYFAMED are performed at 10 m depth using a CARIOCA $pCO_2$ sensor (Merlivat et al., 2018) which monitors
temperature and $pCO_2$ in the seawater covering a range of 200 to 550 µatm. The measurement is based on a
colorimetric method with gas equilibration over a semi-permeable membrane. By means of comparison with the
BOUSSOLE $pCO_2$, discrete carbon samples were collected from DYFAMED in February and March 2020 (Table
124  5).


**2.3      W1M3A and E2M3A fixed stations**
In the Italian EEZ, the open ocean fixed stations W1M3A and E2M3A, which both are part of the ICOS station
network (Steinhoff et al., 2019), are located in the Ligurian Sea and the southern Adriatic, respectively. The
W1M3A, operated by CNR-IAS, consists of a large spar buoy and a subsurface mooring positioned close by
(Canepa et al., 2015). The origins of the observatory date back to the 1970 and the structure was specifically
designed to allow for negligible sensitivity of sea heave and height. The main buoy continuously collects a
complete set of meteorological variables and near surface (0-40 m water depth) hydrographical measurements in
near real time integrated over an hourly basis. On the main pole of the buoy at a nominal depth of 6 m is installed
an instrumental package to monitor $pCO_2$, hydrography, dissolved oxygen, Chl-a, and turbidity consisting of a
CO2-proCV Pro-Oceanus Systems, a SBE16+ equipped with a SBE43 probe by Sea-Bird Electronics Inc., and a
FLNTUS combined fluorometer and turbidimeter by WetLab. The mooring line also provides oceanographic
measurements (temperature and salinity) in delayed mode from the euphotic zone to the dark ocean (about 1000
m depth). Discrete carbon samples were collected from W1M3A in October 2020 (Table 5).
Fixed station E2M3A, operated by OGS, consists of a system with two moorings, the main mooring housing
the surface buoy equipped with a meteorological station and radiometers to collect measurements of air-sea
interaction, sensors for physical (hydrography) and chemical (dissolved oxygen, $pCO_2$ and pH) variables
distributed in the mixed layer, telemetry and services (Bozzano et al., 2013; Ravaioli et al., 2016). The surface
buoy collects the acquired data and transmits them in real time to the online server. On the secondary mooring
line, there is an instrument chain with sensors at different depths for physical and chemical measurements from
the seafloor to the intermediate layer. The station has been in operation almost continuously since 2006 and is the
longest open sea time series in the Adriatic. The E2M3A measures temperature and salinity using a SeaBird SBE
SMP ODO with an integrated optical sensor for dissolved oxygen positioned at 2 m depth.

**2.4      PALOMA and MIRAMARE fixed stations**
The Italian coastal fixed stations PALOMA and MIRAMARE are located in the Gulf of Trieste in the northern
Adriatic Sea and are also part of the ICOS station network. The PALOMA coastal fixed station, operated by CNR-
ISMAR is an elastic beacon located in the centre of the Gulf of Trieste where the water depth is 25 m. The





customised float is designed to provide high stability even with strong winds. This station monitors $p\mathrm{CO_2}$,
dissolved oxygen, and hydrography of the surface water (Ravaioli et al., 2016). The observations are
complemented by meteorological variables, and by monthly ship visits for in situ measurements (CTD profiles
and discrete samples for pH, Total Alkalinity (TA), dissolved oxygen, nutrients) as detailed by Cantoni et al.
(2012). A Contros HydroC sensor is used to determine $p\mathrm{CO_2}$. The measurement principle is based on gas
equilibration across a semipermeable membrane followed by NDIR detection of the $\mathrm{CO_2}$ amount, data were
corrected on the basis of pre and post cruise calibrations (Fietzek et al., 2014). The $\mathrm{CO_2}$ sensor was deployed at 3
m depth and the measurement frequency is normally every 6 hours, but was severely increased when the SDs
encircled the station. The corresponding hydrography and dissolved oxygen were determined using a SeaBird SBE
38-SMP-ODO. Sensors transmit data to shore in near real time. By means of comparing the Contros HydroC $p\mathrm{CO_2}$
measurements, discrete carbon samples were collected from the site on 15 July (Table 5).
The MIRAMARE coastal fixed station, operated by the OGS, is located at the edge of the Miramare Marine
Protected Area and consists of a surface buoy anchored to the seafloor at a depth of 18 m and equipped with a
weather station and sensors for measuring hydrography, dissolved oxygen, $p\mathrm{CO_2}$, and pH. The surface buoy
collects the acquired data and transmits it to shore in near real time. Two additional CTD multiparameter probes
with dissolved oxygen sensors are positioned at 10 and 15 m depth. The buoy has been in operation since January
1999 measuring physical variables, and is, together with biological and chemical time series at the nearby C1 site,
part of the Italian LTER (Long Term Ecological Research) network. At MIRAMARE, a Sea-Bird SBE 37 SMP
ODO with a SBE 63 probe was used for determining hydrography and dissolved oxygen of the surface water. In
addition to the $p\mathrm{CO_2}$ probe described above, a SeaFET pH probe was placed at 2 m depth. On 17 July, discrete
carbon samples were collected from this site (Table 5).

## 2.5 Gliders

In the MOOSE program supported by CNRS, gliders are deployed regularly in the northwestern Mediterranean
basin and in particular along the Nice-Calvi endurance line where the DYFAMED site is located (Bosse et al.,
2015). Ocean gliders sample the ocean along a trajectory between the surface and 1000 m. The typical slope of the
isopycnals is much smaller than the glider's pitch angle, so dives and ascents can be considered vertical profiles
and are separated by typically 2-4 km and 2-4 hours depending on the sampling strategy (Testor et al., 2019).
During this experiment, we used one deployment of the Slocum glider along this endurance line (MOOSE T00-43
mission) performed from 12 March to 20 June 2020. This glider was equipped with a Sea-Bird SBE41CP CTD
probe, an Aanderaa oxygen optode 4330 and a Wet Labs bio-optical fluorometer.
The OGS has established an ocean glider monitoring program in the southern Adriatic since 2014. An ocean
glider performs the Bari-Dubrovnik section twice a year, as part of a multiannual repeated section in the convective
area (Mauri et al. 2016; Pirro et al. 2022; Kokkini et al., 2019). The transect covered during the ATL2MED
demonstration experiment was extended to include the area of the E2M3A fixed station from 12 June to 2 July
2020. During the 20-day campaign 250 dives between 20 to 950 m profiles separated by 3-5 km and 4-6 hours
were collected. The SeaGlider was equipped with a Sea-Bird SBE41CP CTD probe, an Aanderaa oxygen optode
4330 and a Wet Labs bio-optical fluorometer.

## 2.6 Vessel-based research expedition

Discrete samples for Dissolved Inorganic Carbon (DIC) and TA were collected onboard the R/V Meteor during
fall 2019. Furthermore, discrete samples for DIC, TA, pH, and dissolved oxygen are collected regularly in the
vicinity of all of the ocean fixed stations, however, this was not always possible during the ATL2MED
demonstration experiment due to COVID-19 pandemic restrictions. Table 5 is an overview of the discrete samples
collected during the ATL2MED experiment and their sampling depth and analysing methods.

**Table 1. Research vessel and fixed ocean stations from which temperature, salinity and carbon measurements were**
**compared with those of the SDs.**

| Research vessel/ | Position | Institution | SD 1030 | SD 1053 |
|---|---|---|---|---|





| fixed station | | | | |
|---|---|---|---|---|
| R/V Meteor | 17.80°N 20.60°W | GEOMAR (DE) | 30 November 2019 | 12 December 2019 |
| LION | 42.00°N 4.90°E | CNRS (FR) | 1-2 April 2020 | 1-2 April 2020 |
| DYFAMED | 43.42°N 7.87°E | CNRS (FR) | 28 April 2020 | 23 April 2020 |
| W1M3A | 43.83°N 9.12°E | CNR-IAS (IT) | 29 April-2 May 2020 | 28 April-2 May 2020 |
| E2M3A | 41.57°N 18.08°E | OGS (IT) | 29 June-2 July 2020 | 29 June-23 July 2020 |
| PALOMA | 45.62°N 13.57°E | CNR-ISMAR (IT) | 15 July 2020 | 15 July 2020 |
| MIRAMARE | 45.70°N 13.71°E | OGS (IT) | 17 July 2020 | 17 July 2020 |


**Table 2. Harbours and dates of SD maintenance, of which all took place in 2020.**

| Drone \ place | Mindelo (CV) | Telde, Gran Canaria (ES) | Porquerolles (FR) | Imperia (IT) | Cefalù, Sicily (IT) |
|---|---|---|---|---|---|
| SD 1030 | | 12 Feb | 22-23 Apr | | 26 May - 6 Jun |
| SD 1053 | 4-14 Jan | | | 7 May | 26 May - 6 Jun |

**Table 3. Instruments, sensors, accuracy, and associated measurement frequency at the different fixed ocean stations**
**and gliders during the ATL2MED demonstration experiment.**

| Instrument/ sensor | Company/ reference | Variable | Accuracy | Measurement frequency | Used by |
|---|---|---|---|---|---|
| SBE37 | Sea-Bird Electronics, Inc. | T Cond | 0.002°C, 0.0003 S/m | 10 min$^{-1}$ | DYFAMED |
| SBE41 (GPCTD) | Sea-Bird Electronics, Inc. | T Cond | 0.002°C, 0.0003 S/m | 1 s$^{-1}$ | Glider MOOSE T00 |
| SBE-19 | Sea-Bird Electronics, Inc. | T, C | 0.005°C, 0.0005 S/m | 2 day$^{-1}$ | MIRAMARE |
| SBE-16plus v2 | Sea-Bird Electronics, Inc. | T Cond | 0.005°C, 0.0005 S/m | 12 day$^{-1}$ | W1M3A |
| SBE41 (GPCTD) | Sea-Bird Electronics, Inc | T, Cond | 0.002°C, 0.0003 S/m | 1 s$^{-1}$ | Glider South Adriatic |
| SBE37-SMP-ODO | Sea-Bird Electronics, Inc. | T, Cond, O$_2$ | 0.002°C, 0.0003 S/m, > 3 μmol/kg | 15 min$^{-1}$ 60 min$^{-1}$ | PALOMA, MIRAMARE |
| CARIOCA | Merlivat and Brault (1995) | $p$CO$_2$ | 2 μatm | 24 day$^{-1}$ | DYFAMED |



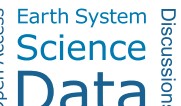

| CO$_2$-proCV | Pro-Oceanus Systems Inc | $p$CO$_2$ | 2 µatm | 12 day$^{-1}$ 6 day$^{-1}$ 24 day$^{-1}$ | W1M3A E2M3A MIRAMARE |
| Contros HydroC | 4H-JENA engineering GmbH | $p$CO$_2$ | 2 µatm | 1 min$^{-1}$ | PALOMA |

T= temperature; Cond=conductivity; O$_2$=dissolved oxygen; $p$CO$_2$=partial pressure of carbon dioxide.
**Table 4. Instruments and sensors at the SDs from Saildrone Inc. during the ATL2MED demonstration experiment**
**and used in this work.**

| Instrument/ sensor | Company/ reference | Variable | Accuracy | Measurement frequency |
|---|---|---|---|---|
| SBE37-SMP-ODO (SD 1030; SD 1053) | Sea-Bird Electronics, Inc. | T, Cond O$_2$ | 0.002°C, 0.0003 S/m, > 3 µmol/kg | 10 min$^{-1}$ |
| ASVCO2 (SD 1030) | PMEL, Sutton et al. (2014) | $p$CO$_2$ | 2 µatm | 24 day$^{-1}$ |

T= temperature; Cond=conductivity; O$_2$=dissolved oxygen; $p$CO$_2$=partial pressure of carbon dioxide.
**Table 5. Instruments and methods used to analyse discrete samples collected at the R/V Meteor and from different**
**fixed stations during the ATL2MED demonstration experiment.**

| Instrument/ sensor | Company/ reference (SOP) | Variable | Accuracy | # measurements (depth) | Facility |
|---|---|---|---|---|---|
| Simultaneous potentiometric acid titration using a closed cell | SNAPO-CO2 prototype, Edmond (1970), Dickson and Goyet (1994) | DIC, TA | ± 2 to 5 µmol kg$^{-1}$ | 1 (5 m) | DYFAMED |
| SOMMA | UiC (SOP 2), Johnson (1992) | DIC | 2 µmol kg$^{-1}$ | 1 (5 m) | GEOMAR |
| VINDTA 3S/VINDTA 3C | MARIANDA (SOP 3b) | TA | 3 µmol kg$^{-1}$ | 1 (5 m) | GEOMAR |
| Hanna Instruments | Titrator HI931 | TA | ± 4 µmol kg$^{-1}$ | 3 (6 m) | W1M3A |
| Automatic potentiometric titrator | Metrohm 685 Dosimat (Hernandez-Aylon, 1999) | TA | 3 µmol kg$^{-1}$ | 5 (0.5, 3 m)[1] | PALOMA |
| Automatic potentiometric titrator | Mettler Toledo G20/SOP3b | TA | ± 4 µmol kg$^{-1}$ | 10 (0.5, 2 m) | MIRAMARE |
| SevenCompact pH metre | Mettler Toledo | pH | ± 0.001 | 3 (6 m) | W1M3A |
| Varian Cary 50 spectrophotometer | Varian, Clayton and Byrne (1993) (SOP | pH | ± 0.003 | 5 (0.5, 3 m)[2] | PALOMA |





| | 6b) | | | | |
|---|---|---|---|---|---|
| Varian Cary 100 Spectrophotometer | Varian, Clayton and Byrne (1993) (SOP 6b) | pH | ± 0.002 | 10 (0.5, 2 m) | MIRAMARE |

O₂=dissolved oxygen; DIC=DIssolved Inorganic Carbon; TA=Total Alkalinity.
[1] For each measurement, 2 replicate samples were collected and analysed.
[2] For each measurement, 2 replicate samples were collected and 2-3 analyses were performed at each replicate.
SOP=Standard Operating Procedure according to Dickson et al. (2007).

**2.7    Argo Float**
Float data were retrieved from the Argo Coriolis Global Data Assembly Center in France (GDAC;
ftp://ftp.ifremer.fr/argo), Wong et al. 2020. For each Argo float the variable SALINITY_ADJUSTED was
extracted, and then used for comparison with SD salinity data. Every profile close in space and time (1 day and 30
km) was chosen and then salinity was averaged in the first 5 metres.

**2.8    Model output**
we used the Copernicus Marine Service (CMEMS) model product, specifically the Global Ocean 1/12° Physics
Analysis and Forecast (https://doi.org/10.48670/moi-00016) and Mediterranean Sea Physics Analysis and Forecast
(Clementi et al., 2021), and daily data was developed for the global ocean and Mediterranean Sea.

**3    Methods**
The ATL2MED demonstration experiment measured water masses with extremely different thermohaline and
biogeochemical characteristics (e.g. the oxygen saturations or phytoplankton blooms) while crossing the Atlantic
Ocean and the Mediterranean Sea.
During the experiment, the two SDs were maintained five times mainly to remove biofouling and to verify
the conditions of the SDs. Despite these interventions, the salinity and dissolved oxygen and Chl-a values measured
by the SDs showed inconsistencies with the average values observed in the adjacent areas by fixed ocean stations,
Argo floats, gliders, satellites and model products. Significant discrepancies were also detected between
measurements of the same variables surveyed by different sensors onboard the SDs. Due to the problems described
above in trying to obtain consistent and reliable data we focus here on the methods used to correct salinity, O₂ and
seawater $pCO_2$ measurements acquired by the sensors mounted on the SDs.
The temperature data acquired by the two SDs were compared with temperature data from the fixed stations
and glides (Table 6) and from these comparisons it became clear that no correction was required for these SD data.

**Table 6. Temperature offsets between SD sensor (SBE37-SMP-ODO) at 0.5 m depth and fixed stations during the**
**ATL2MED demonstration experiment. Green and blue colour refer to SD 1030 and SD 1053, respectively. More**
**details are available in Skjelvan et al. (2021).**

| Fixed station/ glider | Measurement depth (m) | SD 1030 offset (°C) | SD 1053 offset (°C) |
|---|---|---|---|
| W1M3A | 1 | -0.006 | -0.026 |
| E2M3A | 1.7 | 0.216 | 0.138 |
| OGS ocean glider | 0.5 | 0.063 | 0.063 |
| PALOMA | 0.5 | 0.077 | 0.090 |



| PALOMA | 3 | -0.061 | -0.046 |
|---|---|---|---|
| MIRAMARE | 0.5 | -0.085 | -0.205 |
| MIRAMARE | 2 | -0.117 | -0.238 |


### 3.1 Correction of SD salinity data

During the first transect (T1, Fig. 1b), the two salinity sensors on board the SDs showed high consistency (Fig. 2a,
b). After the first maintenance in T2 (Fig. 1b),the SD 1053 showed a

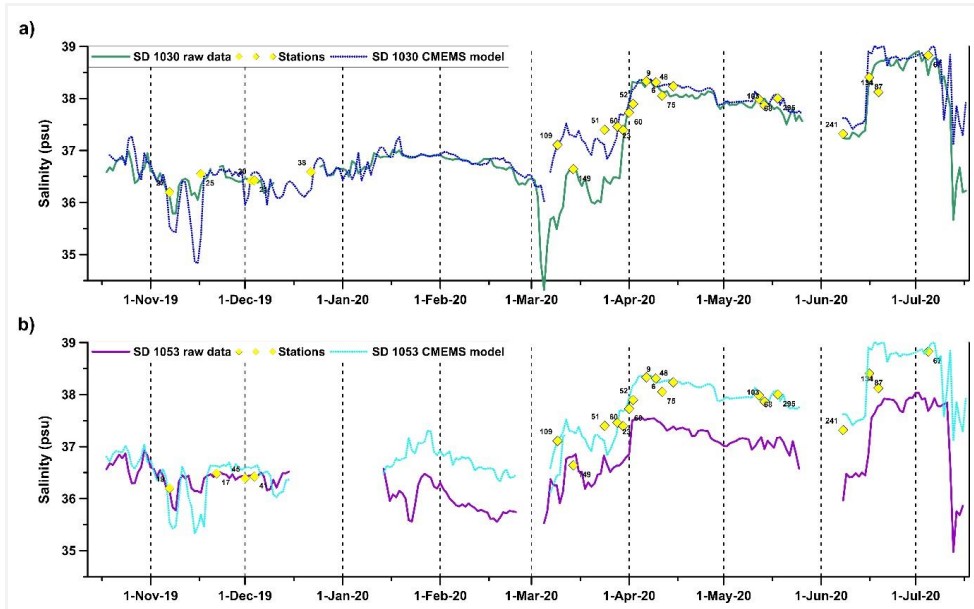

**Figure 2. Salinity time series of daily raw data, CMEMS model salinity products, and float data. (a) SD 1030 salinity raw data (green line), model data from nearest nodes to the SD trajectory (dotted blue line), and Argo float data (yellow diamonds). (b) SD 1053 salinity raw data (purple line), CMEMS model 1053 trajectory (dotted cyan line), and Argo float (yellow diamonds). Numbers close to diamonds indicate the distance (km) from the trajectory of SD.**


reduction in salinity of about 1 compared to the salinity measured by the SD 1030. In T3, the difference in salinity
decreased on average to 0.15. During this period, the SDs crossed the Alboran Sea characterised by high
thermohaline variability due to the presence of Atlantic and Mediterranean waters (Viudez et al., 1998), and the
high spatial and temporal variability in salinity distribution in the area (Cheney and Doblar, 1982) complicates the
understanding of the observed differences (i.e., sensor error or natural variability). In T4 and T5, a salinity shifts
of 1 was observed until the end of the experiment.
Given the large variability found in the salinity data of the SDs, a comparison with *in situ* data along the
trajectory of the experiment is necessary. We first identified the observing systems (fixed buoy, Argo float)
temporally and spatially close to the position of the SDs. Salinity data, with a short temporal and spatial interval
(1 day and 30 km) useful for the comparison and/or correlation were extremely scarce.
To tackle this problem, we decided to compare the data acquired from the SDs with the model products along
the entire route (Fig. 2a, b). The nearest nodes of the model data to the SD trajectory were chosen. An initial
comparison shows good agreement in some parts of the route and some discrepancies which are partly due to a
large time span since the last maintenance. Fig. 2 also shows the distances in km of the SDs from the Argo floats,
which in most cases are quite distant, which on the one hand makes us realise that a regression with floats alone
would be inappropriate. The model, while not deviating much from the float data, can provide us with salinity
products along the SD's trajectory allowing us to correct the salinity trend recorded by the SD. Moreover,
comparative works between the physical model and experimental observations has shown a satisfactory correlation
both in the open ocean (Escudier et al., 2021, Menna et al., 2023) and in the coastal environment (Martellucci et
al., 2021). Despite any limitations a model may have in such cases, we have found the use of model products
ensures a minimum spatial and temporal distance for comparison.
The salinity provided by the model along the two SD trajectories shows a very similar trend to that measured
by SD 1030 with a few exceptions recorded at the beginning of March when SDs crossed Gibilterra strait. In
contrast, SD 1053 showed discrepant values compared to the model and SD 1030, not justifiable with respect to
space-time variability, also SD 1053 showed considerable drift during the experiment.
In the comparison, the area of the Gulf of Trieste in the northernmost Adriatic Sea was excluded due to a very
high spatial and temporal thermohaline variability of the area (Malačič et al., 2006). Moreover, the area of the Gulf
of Trieste is surrounded by three coastlines representing the boundaries of the model , and is not considered to be
useful for the correction.

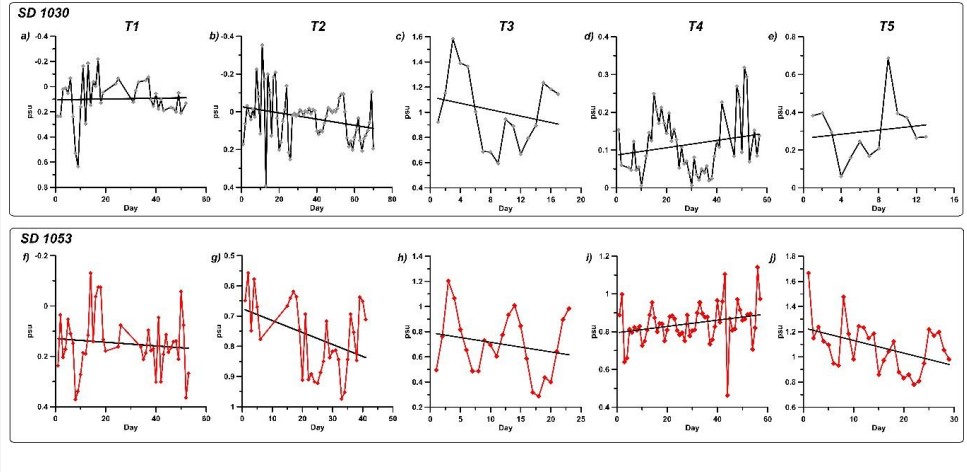

**Figure 3. Differences between model and SD raw salinity during each transect from T1 (left) to T5 (right) for SD 1030 (a, b, c, d, e) and SD 1053(f, g, h, i, j). The x axis is represented as days since maintenance. The solid line represents the linear regression fit.**


Salinity differences between the CMEMS model and the SD 1030 observations shows an underestimation of the
model of 0.1 in T1, which was nearly constant during 60 days of measurements (Fig. 3a), whereas a negative
salinity trend was observed in T2, which corresponds to a 0.1 decrease after 70 days. During the Alboran Sea
crossing (Fig. 3c, T3), the observed salinity deviated strongly from the model (about 0.6) and a consistently high
negative drift was observed over only 20 days, representing an absolute decrease of 0.3. Positive salinity drift was
observed in T4 (absolute decrease of 0.05) across the Ligurian and Tyrrhenian Seas (Fig. 3d, T4), while in the
central Mediterranean Sea (Fig. 3e, T5) the absolute difference was 0.26.
The offset in salinity between the SD 1053 and the model nodes was 0.1, showing a negative difference of
about 0.1 (Fig. 3f). The remaining transects (Figs. 3i-j) showed large offsets between model and observed salinities
(T2: 0.8, T3: 0.7, T4: 0.9, and T5: 1). T2, T3, and T5 show negative salinity differences, 0.15, 0.1, and 0.3,



respectively while T4 shows a positive salinity trend of 1. Given the different drift behaviour of the salinity sensor
(i.e. positive and negative trend, Figs. 3a-e), we decided to correct the different transects separately by applying
variable time offsets to correct both offset and drift.
The salinity correction method was as follows: Daily averages of SD salinities were compared to model
salinities, and the differences were used to estimate the drift and the offset. These variables were obtained using
the least squares minimization method (Figs. 3a-e), with days before maintenance as a time variable. The corrected
SD salinities were compared with the discrete data from the fixed ocean stations.

**3.2     Correction of SD dissolved oxygen data**
Due to the strong dependence of dissolved oxygen on temperature we first analyse the temperature along the track
of the SDs. During the experiment, sea temperature (Fig. 4a) showed a seasonal signal similar to those observed
at these latitudes. The high observed temperature variability also includes the wide geographical coverage of the
SDs. The highest temperatures were measured in November 2019 and July 2020 in the tropical Atlantic and the
southern Adriatic, respectively. The lowest temperatures were measured in the Gulf of Lion in April 2020. Along
the SD tracks, the salinity (Fig. 2b) showed a gradual increase from the Atlantic Ocean to the Eastern
Mediterranean Sea.
Given the correct temperature measurement, any dissolved oxygen drift can be assessed through comparison
with saturation values, this procedure was also used to correct Argo float data with climatological observations
(Takeshita et al. 2013). The percent dissolved oxygen saturation (Fig. 4b) showed a gradual decrease from 100%
at the start of the demonstration experiment to 80% at the end. This behaviour is also reflected in the dissolved
oxygen concentration, which decreases by about 60 µmol/l for SD 1030 (Fig. 4c) and 130 µmol/l for SD 1053
(Fig. 4d) over the course of nine months and the standard deviation of the all uncorrected oxygen record of 16.39
µmol/l and 73.344 µmol/l for SDs 1030 and 1053 respectively. The low correlation ($R^2$ about 0.6) suggests that
the drift observed in dissolved oxygen saturation is independent of temperature variations in oxygen, as seen in
Fig. 4e and 13.5 °C in Fig. 4f, where large ranges of oxygen values at 24 °C , but is more likely due to sensor-
related problems, as was also observed for salinity.
To evaluate the response of dissolved oxygen variation during the demonstration experiment, the 50 days
mean (yellow line in Fig. 4c and blue line in Fig. 4d) was subtracted from the oxygen time series. The residuals
time series (Figs. 4g-h) highlights the same oscillations for both SDs, ranging from 40 µmol/l to -40 µmo/l and do
not show significant trend in sensor response. Prior to applying correction all the unreasonable oxygen
measurements were excluded.

**Figure 4. Time series of hourly (a) sea temperature (b) dissolved oxygen saturation (c-d) and dissolved oxygen from SD 1030 (black line) and SD 1053 (red line), yellow and blue line represent the 50 days mean. (e-f) Comparison**

> **between dissolved oxygen concentration and sea temperature from SD 1030 (black dots) and SD 1053 (red dots). (g-h) Residual from dissolved oxygen (raw data minus 50 days mean).**

After the first analysis we proceeded to correct the negative trend, using the same oxygen correction method as
used in the Argo program (Bittig et al., 2018). The principle of this method is to compare the oxygen measurements
performed while the Argo oxygen sensor is in air with the oxygen partial pressure ($pO_2$) in air (Johnson et al.,
2015). The latter variable is easily calculated from air temperature, air pressure, and relative humidity acquired by
the SDs. Considering that the SD oxygen sensor is installed on the hull about 0.5 m below sea surface and that the
SDs sailing cause mixing of the water surface while sailing, we assume that the SDs oxygen sensors were in
equilibrium with the atmosphere above, and furthermore, we can correct for the oxygen sensor drift using the in
air calibration method (Bittig et al., 2018; Johnson et al., 2015). Specifically, we computed vapour pressure ($V_p$)
from the empirical equation reported in the operating manual of Aanderaa oxygen optode (model 4330) using the
air temperature ($T_{sd}$) recorded from SDs:

$$V_p = e^{\left(52.57-\left(\frac{6690.90}{T_{sd}+273.15}\right)\right)-4.681*log^{T_{sd}+273.15}}$$

and expected partial pressure ($E_{PP}$) from volume fraction of oxygen ($V_{fO2}$; Glueckauf, 1951), atmospheric pressure
($AP_{sd}$ ), vapour pressure ($V_p$) and relative humidity ($RH_{sd}$), as follows:

$$E_{PP} = V_{fO2} * \left(AP_{sd} - \left(V_p * \frac{RH_{sd}}{100}\right)\right)$$

The $E_{PP}$ were then compared to the $pO_2$ from the SDs to compute the gain (G) for daily correction.

$$G = \frac{E_{PP}}{pO2_{sd}}$$

The corrected oxygen data from the SDs ($SD_{csd}$) was calculated from adjusting the oxygen data from SDs ($O2_{sd}$)
with the gain .

$$O2_{csd} = G * O2_{sd}$$

The gain factor was multiplied by the hourly oxygen data allowing to correct the negative trend and the
temperature dependence, thus obtaining a detrended time series.
After detrending the percent dissolved oxygen saturation (Fig. 5a), the saturation values ranged between 99%
and 103%. After detrending dissolved oxygen concentration, the inverse correlation (Figs 5b-c) with temperature
increases significantly ($R^2$=0.98). This results would be expected in a surface layer (in near equilibrium with the
atmosphere) in different seasons (i.e., high oxygen saturations in the warm season and low in the cold season)
without considering biological activity. This procedure using the relationship between oxygen in atmosphere and
ocean allows to correct the trend, as observed for oxygen saturation time series. However, to correct the oxygen
measurements, the biological production and consumption need to be taken into account. To overcome this
problem, the residuals were added to the detrended time series (Figs. 4g-h), considering the residuals as the effect
of biological activity.

**Figure 5. (a)** Time series of detrended (SD 1030 black line and SD 1053 red line) percent dissolved oxygen saturation. Comparison between SD sea temperature and **(b-c)** detrended dissolved oxygen concentration and **(d-e)** corrected dissolved oxygen concentration, where back dots refer to SD 1030 and red dots to SD 1053. Time series of **(f)** corrected dissolved oxygen concentration and **(g)** corrected percent dissolved oxygen saturation.





The comparison between corrected dissolved oxygen and ocean temperature (Figs. 5c-d) shows a lower correlation
(0.91 respect to 0.98) with respect to the detrended time series, suggesting that this lower correlation can be due
to the effect of biological activity. The corrected oxygen measurements (Fig. 5f) spans from 190 µmol/l to 280
µmol/l highlighting the highest concentrations during the spring 2020. Time series of percent dissolved oxygen
saturation does not show any significant trend. Particularly interesting were the periods characterised by
oversaturation (at the end of October 2019, ~120% and at the beginning of March 2020, ~ 110%), and
undersaturation (at 1-2 of April 2020, ~95% and 8-11 July 2020, ~92%).

### 3.3 Comparison with $p$CO$_2$ data at the fixed ocean stations

The sensor $p$CO$_2$ measurements from the different fixed ocean stations are regularly compared to the $p$CO$_2$
calculated from discrete water samples collected by the fixed stations and analysed for TA, pH, and DIC. During
the last half of the ATL2MED, this routine was hampered due to COVID-19 restrictions, thus, between March and
July 2020, there were less discrete carbon samples for comparison with sensor $pCO_2$. Furthermore, there was minor
variability in sampling frequency with regards to the sensor $pCO_2$ measurements and in the pair of measured
variables used for $p$CO$_2$ calculation (TA-pH, DIC-pH, or DIC-TA) between the different fixed ocean stations.
During the ATL2MED demonstration experiment, DIC, TA, and pH were analysed according to SOP 2, 3b, and
6b, respectively (Dickson et al., 2007) with some minor local variations (Table 5). Certified Reference Material
(CRM) and TRIS provided by Prof. A. Dickson (Scripps, USDC, USA) were used to determine the accuracy. $p$CO$_2$
was calculated using the speciation software CO2SYS (Pelletier et al., 2007), with the discrete carbon pairs DIC-
TA or TA-pH as input variables. In the computation, the carbonate system constants from Lueker et al. (2000), the
HSO$_4^-$ constant from Dickson (1990), the total borate-salinity relationship of Lee et al. (2010), and the KF constant
from Perez and Fraga (1987) were used. There are uncertainties connected to the pCO$_2$ calculation and thus, no
adjustments were performed for the station pCO$_2$ sensor data when the deviation between the pCO$_2$ acquired by
the station sensors and calculated from discrete carbon data were less than 10 µatm and 7.5 µatm for the discrete
carbon pairs DIC-TA and pH-TA, respectively. Uncertainty thresholds were set based on measurement
uncertainties at each facility and temperature and pCO$_2$ in the vicinity of the fixed stations. On this basis, none of
the stations' pCO$_2$ sensor data needed to be adjusted.

### 3.4 Correction of SD CO$_2$ data
The general accuracy of the ASVCO2 system attached to the SD 1030 was checked by PMEL prior to deployment
by comparing the results with ESRL CO$_2$ standards traceable to WMO standards (Sutton et al., 2014). For this test,
typically 6 standard gases were used. On the return of the ASVCO2 system to PMEL, it was discovered that the
span gas was adjusted too low to completely flush the detector and that this had been so during the whole
ATL2MED demonstration experiment. Thus, the LI-COR had to be recalibrated at the PMEL lab and this implied
that the onboard gas spanning was bypassed and new calibration coefficients were developed. Furthermore, the
pre-mission test data from the PMEL lab were reprocessed using the new calibration coefficients. Based on the
reported issues with the ASVCO2 instrument, the accuracy of the CO$_2$ measurements is estimated to be < 5 µatm.
$p$CO$_2$ and $f$CO$_2$ (µatm) from the ASVCO2 instrument were calculated according to Sutton et al. (2014) using
T and S from the SBE 37-SMP-ODO attached to the SD 1030. Fig. 6a shows the uncorrected and corrected $p$CO$_2$
acquired from the SD 1030. In Fig. 6b, the difference between corrected and uncorrected $p$CO$_2$ is shown and the
offset increases from approximately 1 µatm at the start of the experiment to approximately 12 µatm at the end.

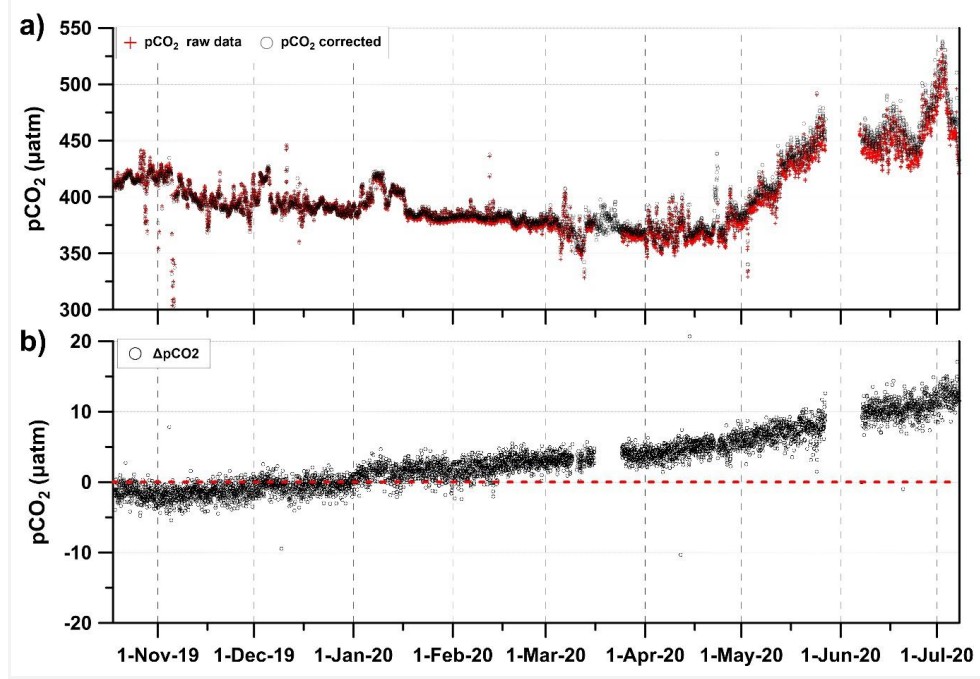

**Figure 6. a): Raw (red crosses) and corrected (black circles) $p$CO$_2$ data acquired by the SD 1030 as functions of time. (b): $\Delta$CO$_2$ (corrected $p$CO$_2$ - raw $p$CO$_2$).**


## 4    Results and discussion

### 4.1    Salinity

The corrected salinity for both SDs showed very similar values (Figs. 7a-b), with the difference between raw and corrected salinity values increasing with time. Overall, the correction was largest for SD 1053 (Fig. 7b). During the first part of the mission (T1) no significant difference appears between raw and corrected salinity data, the salinity corrected SD data was fairly similar to the CMEMS model. To validate the salinity corrected data a comparison with different observing systems was done (Figs. 7c-i). During the first part of the mission (i.e. T1) the corrected salinity for SD 1053 overestimated (~0.15) the salinity values (Figs. 7c-d), the raw salinity instead seems to be in agreement with respect to Argo float data, highlighting differences less than 0.01.




**Figure 7. (a) Salinities from SD 1030 (corrected black line and raw data green line), and the CMEMS model products (violet line). (b) Salinities from the SD 1053, (corrected red line and raw data magenta line), and the CMEMS model products (cyan line). (c-f) Plot of SD corrected data (1053 red triangle and 1030 black triangle) SD raw data (1053 magenta triangle and 1030 green triangle) and Argo float (yellow diamonds) over the SD trajectory. The labels close to the diamonds indicate the distance (km) from the SD trajectory. Comparison (g) among SD 1053 (red triangle), SD 1030 (black triangle) and ocean glider (yellow dot, MOOSE program) in the Gulf of Lion; (h) among SD 1053 (red), SD 1030 (black), E2M3A (blue line) and ocean glider (yellow diamond, OGS-program) in the Southern Adriatic; and (f) among SD 1053 (red),SD 1030 (black) and MIRAMARE station in the Northern Adriatic.**

The corrected salinity data for SD 1030 showed a slight overestimation of salinity, while the raw salinity data
showed an underestimation. The SD 1030 salinity corrected highlights good agreement in T1 respect to the SD
1053, the average difference was less than 0.05, the highest difference between Argo float data and corrected
salinity data can be observed on 17 November 2019 (~0.15). In T2, the comparison can only be made for SD 1030
with only one Argo float profile (Fig. 7e), the corrected salinity differs by about 0.02, while the raw data show a
large difference of about 0.05. The corrected values seem to be in agreement with the observations of the Argo
floats during the crossing of the Mediterranean Sea (Fig. 7f). The average difference between the corrected salinity
and the Argo float data was less than 0.05, while the difference in the raw data was about 0.65. In the southern
Adriatic, the SDs spent four days sampling the area, which allowed a robust comparison between E2M3A fixed
ocean station and the glider measurements (Fig. 7h). The comparison showed a very good agreement between the
observations, which had almost the same salinity. In the northern Adriatic (Fig. 7i), the comparison showed poor
results in terms of absolute salinity values. However, the three time series of salinity (Fig. 7i) showed the same
trend with an average difference between the SDs and the Miramare fixed ocean station of ~0.3. The major
differences between the two SDs are mainly due to their temporal and spatial distance (> 100 km around 1 June in
Fig. 7b; 20 km on 27 April in Fig. 7h; ~25 km on 26-27 July in Fig. 7i).
The comparison with the different fixed ocean stations shows that the corrected salinity in T2, T3, T4 and T5
is consistent with the values measured at the stations (Argo float, glider and buoy), the differences are mainly due
to the distance between the different observatories and to the natural variability of the areas. Considering that in
T1 the SDs raw data showed a smaller deviation from the Argo float data, we decided to apply the salinity
correction after 10/01/2020 corresponding to the start of T2.

## 4.2    Dissolved oxygen

Due to COVID-19 restrictions, none discrete dissolved oxygen measurements were available over the period of
ATL2MED demonstration experiment, thus, the SD corrected dissolved oxygen cannot be compared with such
measurements. For these reasons we evaluated the change in dissolved oxygen measured by the two SDs in two
different geographical areas, where oxygen showed oversaturation (Fig. 8) and undersaturation (Fig. 9). To do
this, we compared Chl-a and temperature data to evaluate if the correction could be reasonable or not with
ecosystem dynamics.
The oxygen saturation concentration can be expressed as a function of salinity and temperature, in terms of
solubility. The gas concentration in seawater depends on thermohaline characteristics and biological activity. The
solubility of oxygen decreases with increases in temperature and salinity, showing a strong linear correlation. In
the ocean, oxygen saturation slightly lower than 100% can be observed during the cold seasons while in the warm
season oxygen saturation is slightly higher than 100%, inversely to the oxygen concentrations (i.e., high
concentrations during cold season and low in the warm season). This is because heating and cooling are faster than
outgassing. Furthermore, oxygen concentrations are affected by primary production, which adds to the oxygen
oversaturated surface water during the productive season. In the same part of the ocean oxygen saturation can
reach values greater than 110% strongly related to biological production, moreover it is possible to observe
saturations less than 90% strongly related to the biological consumption and advection of deep water, poor in
oxygen.
In order to test the validity of the correction made, it was chosen to evaluate the oxygen measurements in the
events in which the sensor measured oversaturation and under saturation. To evaluate the ocean response, sea
surface Chl-a (OCEANCOLOUR_MED_BGC_L3_NRT_009_141, doi: 10.48670/moi-00297), sea surface
temperature (SST_MED_SST_L4_NRT_OBSERVATIONS_010_004, doi:10.48670/moi-00172) and the vertical
structure          of          ocean          temperature          (MEDSEA_MULTIYEAR_PHY_006_004,
doi:10.25423/CMCC/MEDSEA_MULTIYEAR_PHY_006_004_E3R1) were downloaded from the CMEMS data
portal and analysed.
Between 25 and 29 October, the dissolved oxygen concentration and saturation around the Canary Islands
was high (>240 μmol/l and >110%; Fig. 8a and b). In the same period the SDs measured high concentrations of
Chl-a (~ 2 μg/l, Fig. 8b, blue and orange line). The optical sensors at the SDs and thus, the Chl-a measurements,
were strongly affected by biofouling for most of the demonstration experiment, which is why we in general do not
use to these measurements in this work. However, during the 10 first days in October 2019, the Chl-a data acquired
by the SDs seemed to produce reasonable values, as for a new sensor the increase in biofouling need weeks to
become significant (Delory et al., 2018), and thus, we refer to these Chl-a data, collected by the SDs in the transect
T1, when explaining the dissolved oxygen oversaturation episode off the Canary Islands. The patch of high Chl-a
was also evident in the satellite images of sea surface Chl-a (Fig. 8c) and sea surface temperature (Fig. 8d). High
Chl-a concentrations and low temperatures identify a circulation structure that has moved away from the African
coast. Considering that the latter is a very productive area due to the permanent upwelling off NW Africa coast
(Cropper et al., 2014; Fischer et al., 2016), this justify the high Chl-a concentration observed by the SDs at that
time.

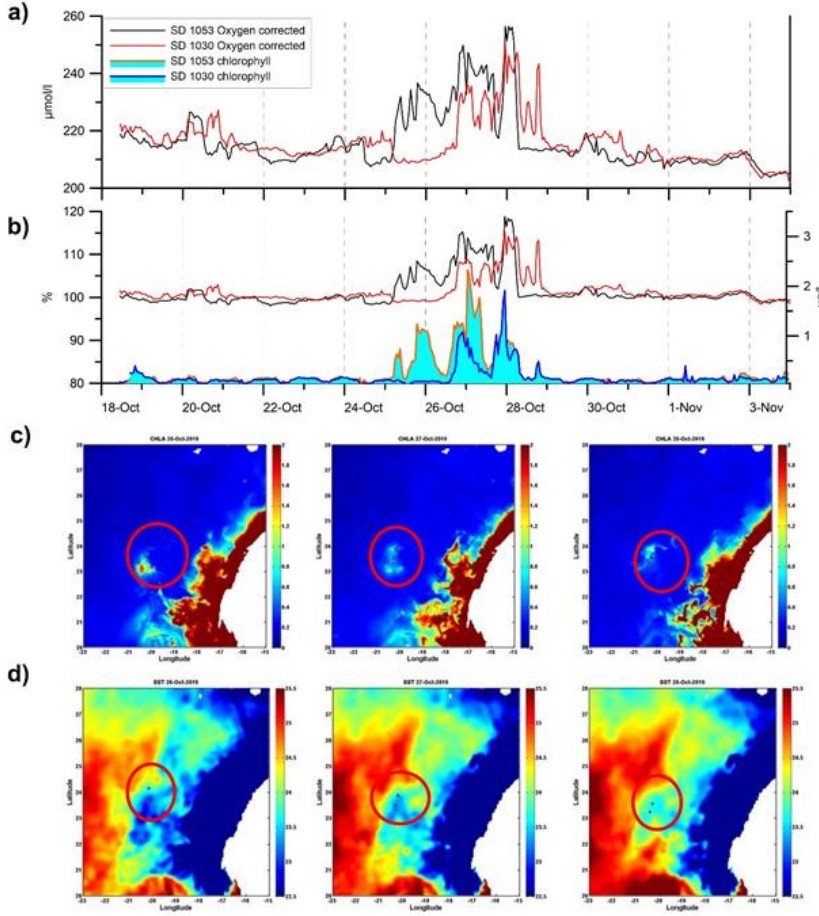

**Figure 8. Time series of (a) dissolved oxygen concentration and (b) percent dissolved oxygen saturation in the Canary**
**Islands area. Evolution of (c) sea surface Chl-a and (d) sea surface temperature between 26 and 28 October 2019, the**
**red circle identifies the position of SDs.**



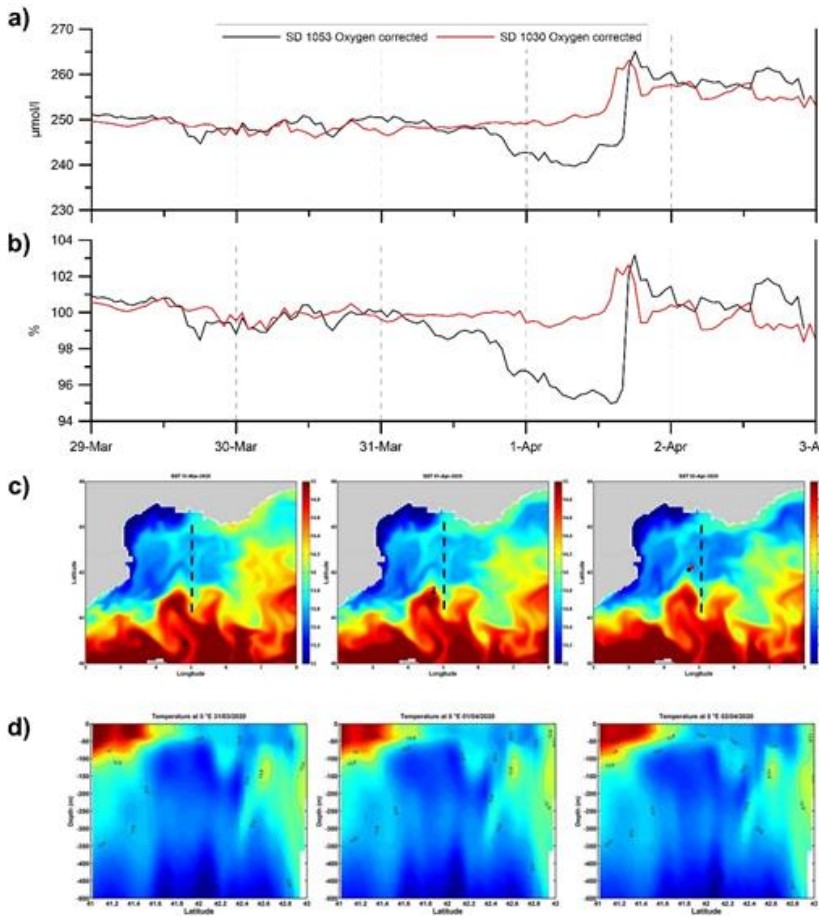

**Figure 9. Time series of (a) dissolved oxygen concentration and (b) percent dissolved oxygen saturation in the Balearic**
**basin. (c) Sea surface temperature evolution between 31 March and 2 April 2020. The black dotted line highlights the**
**vertical section in (d).**
Between 29 March and 3 April 2020, the SDs crossed the Balearic basin and on 1 April, 2020, they measured a
decrease in dissolved oxygen concentrations of about 10 umol/l (Fig. 9a). This behaviour was also observed in the
percent dissolved oxygen saturation (Fig. 9b) which reached values lower than 95%. The northern part of the basin
was characterised by the presence of low temperature at surface (Fig. 9c) with respect to the southern part. The
vertical temperature section (Fig. 9d) highlights the presence of a strong upwelling, that brings cold water through
the surface justifying the cold water observed in Fig. 9c. The presence of this upwelled water caused the decrease
in oxygen saturation (Fig. 9b) observed by SDs, as the upwelled water is commonly characterised by low oxygen
concentrations due to the biological respiration.

**4.3    $p$CO$_2$**
The $p$CO$_2$ sensors at the different fixed stations were deployed at depths between 2 to 10 m while the SD measured
at 0.5 m depth. To be able to compare $p$CO$_2$ measurements from the different depths, the station $p$CO$_2$ data were
normalised to surface temperature by using the relationship of Takahashi et al. (1993): $pCO_2(1) =$
$pCO_2(2)\,exp^{0.0423(T_1-T_2)}$, where T is temperature and 1 and 2 refer to the measurements at 0.5 m depth of the SD
and at the measurement depth of each local station, respectively. Furthermore, the $p$CO$_2$ measurements acquired

by the SD 1030 were compared to the corrected $p$CO$_2$, surface temperature normalised, from the fixed ocean
stations (Fig. 10 and Table 7). The difference varied between -0.5 and -16.9 µatm. The largest difference occurred
in the Eastern Atlantic, where calculated $p$CO$_2$ from discrete DIC and TA where compared to the SD 1030 data.
Part of this deviation is likely attributed to calculation errors which is estimated to be about 10 µatm when errors
in both DIC, TA, and the carbon constants are included (Orr et al., 2018). The smallest difference between the SD
1030 $p$CO$_2$ and the fixed ocean stations are seen at DYFAMED toward the end of April 2020 (-2.9 µatm) and at
MIRAMARE in mid July 2020 (-0.5 µatm). The larger discrepancy at W1M3A and PALOMA might be attributed
to processes which are not taken into account by temperature normalising, e.g., spatial gradients due to primary
production/remineralization, which would decrease/increase the $p$CO$_2$.

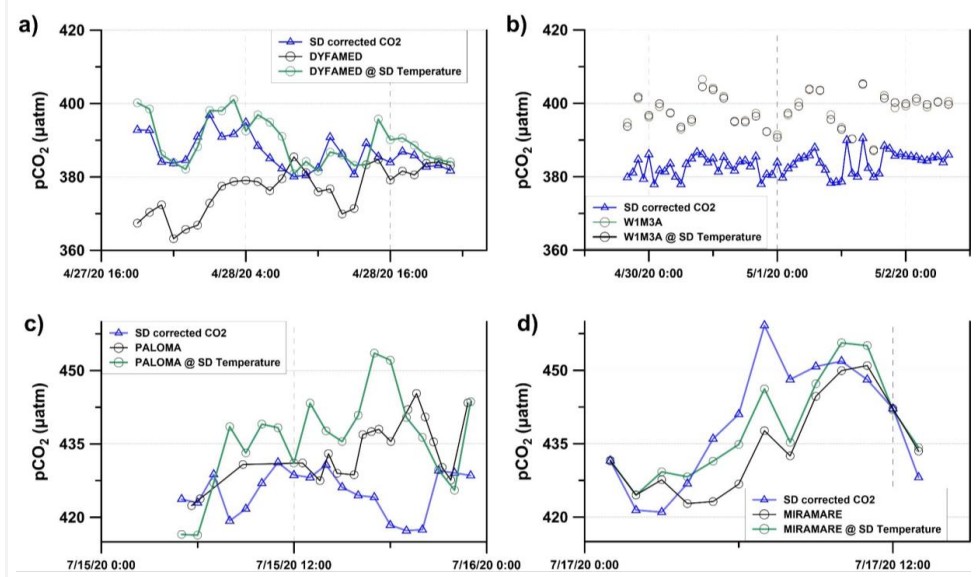

**Figure 10. Comparison between $f$CO$_2$ measured by the SD 1030 (blue) and at the fixed ocean stations (black and
green): the fixed ocean station $f$CO$_2$ measured at in situ depth and temperature (black) and the fixed ocean station
$f$CO$_2$ normalised to surface temperature (green).**

**Table 7. Comparison between pCO$_2$ measurements at SD 1030 and the fixed ocean stations.**

| Station/ platform | Measurements | Date | Deviation between $p$CO$_2$ at SD 1030 and $p$CO$_2$ at fixed station normalised to SST (µatm) |
|---|---|---|---|
| R/V Meteor | Discrete samples @ 5 m | 30 Nov 2019 | -16.9 µatm |
| DYFAMED | CO$_2$ sensor @ 10 m | 27-28 Apr 2020 | -2.9 µatm |
| W1M3A | CO$_2$ sensor @ 8 m | 28 Apr - 2 May 2020 | -14.2 µatm |
| PALOMA | CO$_2$ sensor @ 3 m | 15 July 2020 | -14.7 µatm |
| MIRAMARE | CO$_2$ sensor @ 2 m | 17 July 2020 | -0.5 µatm |

## 5 Data availability





The data used in this work comes from a variety of sources (SDs, fixed ocean stations, gliders, Argo floats, and
model product) and Table 8 is an overview of where to find the data.
**Table 8. Data availability**

| Data creator | Variables used in current work | Doi |
|---|---|---|
| SD 1030 | T, S, DO, pCO$_2$ | https://fileshare.icos-cp.eu/s/eyLp9m685QA8ME7; Skjelvan et al. (2023) |
| SD 1053 | T, S, DO | https://fileshare.icos-cp.eu/s/eyLp9m685QA8ME7; Skjelvan et al. (2023) |
| R/V Meteor | T, S, DIC, TA | https://fileshare.icos-cp.eu/s/eyLp9m685QA8ME7; Paulsen et al. (2023) |
| DYFAMED/BOUSSOLE fixed station | T, S, DIC, TA, pCO$_2$ | https://doi.org/10.17882/43749; Coppola et al. (2023) |
| Nice - Calvi glider | S | https://www.seanoe.org/data/00409/52027/ Testor et al. (2017). doi from the MOOS program (glider SLOCUM Theque on MOOSE T00_43 section) |
| W1M3A fixed station | T, S, pCO$_2$ | https://fileshare.icos-cp.eu/s/eyLp9m685QA8ME7; Bozzano and Pensieri (2023) |
| E2M3A fixed station | T, S, pCO$_2$ | https://nodc.ogs.it/catalogs/doidetails?4&doi=10.6092/d0d50095-bd30-4ff7-8d0a-a12121e72f78; Cardin et al. (2020) |
| E2M3A glider | S | https://nodc.ogs.it/catalogs/doidetails?8&doi=10.13120/e7277c6b-444a-4d61-8288-596af1bac3ff ; Gerin et al. (2021) |
| PALOMA fixed station | T, S, pH, TA, pCO$_2$ | https://fileshare.icos-cp.eu/s/eyLp9m685QA8ME7; Lucchetta and Cantoni (2023) |
| MIRAMARE fixed station | T, S, pH, TA, pCO$_2$ | https://fileshare.icos-cp.eu/s/eyLp9m685QA8ME7; Giani (2023) |
| Argo buoy | S | https://doi.org/10.48670/moi-00044; Wong et al. (2020) |
| CMEMS | Model product | https://doi.org/10.25423/CMCC/MEDSEA_ANALYSISFORECAST_PHY_006_013_EAS7; Clementi et al. (2021) |


**6    Summary**

The ATL2MED demonstration experiment, which lasted for 273 days, represented the first monitoring
experiments of SDs covering both in the Eastern Tropical North Atlantic and the Mediterranean Sea, evaluating
dynamics between fixed ocean stations within the same basin as well as comparing characteristics between basins.
The experiment covered all seasons with varying meteorological and oceanographic conditions, primary
productivity, and maritime traffic. The ATL2MED lasted longer than planned primarily due to challenges with





heavy biofouling at the hull of the two SDs, COVID-19 pandemic restrictions, low winds, and strong contrary
winds.
A huge amount of data covering a large area from the eastern North Atlantic to the northern central
Mediterranean has been produced, which required quality control and assurance to a varying degree, primarily
depending on how sensitive the sensors were to biofouling. Due to the COVID-19 pandemic restrictions, there
was a lack of validation samples collected from cruise transects, and this has enforced a new way of thinking
regarding drift correction. The SBE salinity sensors at the SDs have been corrected, when necessary, using model
products and the method was validated by comparing the data corrected with available in situ measurements. The
Aanderaa dissolved oxygen sensors at the SDs were corrected using a new method that uses the in air oxygen
measurements to correct the trend. The corrected SD datasets fit well with data from fixed stations and gliders,
which means that the correction methods used are valid. The output is data sets which are available for process
interpretations in future research.
One limitation of the correction methods has been the limited amount of validation points like Argo floats,
fixed stations, cruises etc. This has been solved by taking model products into consideration. Notwithstanding the
limitations of the method for correcting the data the use of the model ensures remarkable consistency, guaranteeing
consistent correction in both space and time.
Other SD sensors were affected by biofouling to such a degree that the datasets were unable to correct given
the limited samples for validation, like e.g. the optical sensors for fluorescence measurements. Some
recommendations connected to this are put forward in the next section.

## 7 Future and recommendations

The ATL2MED demonstration experiment is a nice example of how ASV can be used to perform multi-variable
and high-resolution sampling from areas which are not easily accessible, e.g. due to remote location, limited
shiptime availability, or COVID-19 restrictions. The SDs are environmentally friendly platforms, and they,
together with other ASVs, are useful as a complement in the validation of fixed ocean stations. However, the
experiment clearly shows some of the challenges faced when this type of surface vehicle is part of long-term
missions. In general, the use of SDs requires a severe amount of effort into securing that the data are of
scientifically usable quality. More specifically, the sensors installed on the SDs always remain in the surface layer
and are exposed for biofouling, which can be particularly impacting in relatively warm waters of the Mediterranean
Sea, and not only during summer. For future experiment, a maintenance and sensor cleaning frequency depending
on the area should be implemented. In situations where this is not possible, biolimiting equipment should be used,
like UV systems powered by the solar panels and wipers which regularly clean the optical sensors. Furthermore,
regular cleaning of the hull will also ensure the necessary manoeuvrability and navigation precision. Experiences
from the ATL2MED demonstration experiment showed that the RBR sensor package used at the SDs had serious
issues regarding the biofouling effect. After 9 months in sea, this is somewhat expected. However, the SBE37
sensors seem to be more reliable and robust regarding biofouling, but a regular sensor cleaning procedure is
necessary using special devices or human interventions during the SD deployment. Regarding correction of
dissolved oxygen, it is advised to facilitate an in air calibration like the one used for Argo floats. This would require
some reorganisation of the sensors, however, it will be easier to correct for drift of the oxygen sensor. It is also
advised to look into the location of the SD sensors. For instance, the RBR sensor at the SD 1053 measured
significantly lower dissolved oxygen compared to the SBE. One possible explanation for this could be that the
RBR sensor was mounted inside the ship keel where dead water could accelerate the sensor fouling. The sensor
mounting must ensure that the SD sensors are mounted correctly to sample open water.
The ATL2MED demonstration experiment suffered by lack of discrete samples for validation. Thus, future
experiments should be organised in such a way that discrete samples for salinity, oxygen, carbon, and chlorophyll-
a are collected at a reasonable frequency, which will ease the validation of the SD dataset quality tremendously.
Finally, the capability of the Saildrone vehicles as tools for validating other types of measuring devices (e.g., fixed
ocean stations, mobile platforms or ships) strongly depends on several conditions such as distance from the other
platforms, depth of fixed station measurements, environmental conditions and status of the sensors.



**Competing interests.** None
**Acknowledgement.** The ATL2MED experiment has received generous funding from the US company PEAK 6
Invest and invaluable support regarding coordination, operation, and data deliverance from Saildrone Inc..
Furthermore, funding has been provided by GEOMAR Helmholtz Centre for Ocean Research (GEOMAR),
Integrated Carbon Observation System - Ocean Thematic Centre (ICOS-OTC), the French National Centre for
Scientific Research (CNRS), Oceanography Laboratory of Villefranche (LOV), the Oceanic Platform of the
Canary Islands (PLOCAN), Ocean Science Centre Mindelo (OSCM), the Hydrographic Institute of Portugal (IH),
Balearic Islands Coastal Observing and Forecasting System (SOCIB), Italian National Institute of Oceanography
and Applied Geophysics (OGS), Helmholtz Zentrum Geesthacht (HZG), Centre Scientifique de Monaco (CSM),
National Research Council-Institute of Marine Sciences (CNR-ISMAR), and National Research Council - Institute
for the study of Anthropic Impact and Sustainability in the Marine Environment (CNR-IAS). We thank the OGS
engineers Paolo Mansutti and Giuseppe Siena for the assistance during the final recovery of the SDs, and Piero
Zuppelli, Riccardo Gerin, Antonio Bussani and Massimo Pacciaroni for piloting the OGS glider. Furthermore, we
thank Björn Fiedeler and Benjamin Pfeil for initialising the demonstration experiment and for executing the first
phase of the experiment. Finally, we thank Adrienne Sutton and Stacy Manner for invaluable help with correcting
the ASVCO2 $pCO_2$ data.

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
