# Peer review of "CO2 and hydrography acquired by Autonomous Surface"

_Earth System Science Data, 2023_

## Referee Comment (RC2)

**Review for Manuscript essd-2023-457 - CO₂ and hydrography acquired by Autonomous Surface Vehicles from the Atlantic Ocean to the Mediterranean Sea: data correction and validation – Martellucci *et al.***

*\*\* General \*\**

The paper by Martellucci et al. presents a study of the correction applied to several biogeochemical parameters acquired by two SailDrone platforms during the Atlan2Med demonstration experiment. The two platforms sailed from the Eastern Tropical Atlantic Region to the North-Western Mediterranean Sea between July 2019 and October 2020. The authors have put together an interesting new dataset and reviewed choices regarding how and when data adjustments were done. Their findings show significant offsets between raw data and reference datasets from nearby cruises and stations. They arrive at a plausible set of conclusions and suggest useful corrections. The resulting paper could be a useful contribution to the literature as it is needed for deeper research, but I believe it should be returned to the authors for deep revisions and changes, as explained below.

*\*\* Major comments \*\**

1. The manuscript has been submitted as an original research paper, but its content is rather technical and descriptive, and the discussion is quantitative rather than qualitative, and did "feel" technical. I recommend submitting the paper as a technical note, even if in its actual form, this is too long to be accepted in this format. The paper should focus solely on detailing the data qualification and correction, omitting particularly the site descriptions (already published), several figures, and tables that could be either completely removed or put to the supplementary material. This paper needs to be shortened and focused. In its current form (without real comparison or added-value), the paper does not meet the publication criteria.

2. My main concern in this paper is the corrections presented in Section 3 and the lack of robust statistics. It is not clear what hypothesis has been tested, nor which test is used. The authors need to include several other statistics (such as p-value, and RMSE) for their regressions. There is also no detail on how the regressions were determined (ordinary least squares I assume, but why is that appropriate for some parameters such as the temperature that has so few data points?). In Section 3.2 the authors claim several times that oxygen data have either a temperature dependence or independence and seem to construct the entire correction based on this observation (not statistically proven). It really does not look like that in Figure 4. This claim needs to be substantiated, and the methods used to determine this evaluation detailed. Based on the very little information given on methods and assumptions I do not believe these results are valid… and that leads me to question the analysis done using Argo data.

3. The quantitative aspects seemed, in places, potentially incorrect (i.e., the temperature and salinity data analysis). Comparisons with the literature are missing in a lot of places, leading to a confusing feeling about the accuracy of the corrected datasets. In other places, comparisons against data are done while it was stated previously that the data were either incorrect (Chl-a data) or acquired at a distance too far from the SD track (float data). A comparison with historical datasets would be a way forward to reduce this feeling of an incomplete and somehow superficial analysis. Generally, several references are missing, in both the discussion section and in the introduction to better describe the studied area and its characteristics.

4. Some of the figures are missing information, some others could be split and simplified, and the writing is difficult to understand in some places (but excellent in other parts). Some of the notation is inconsistent (see line-by-line comments). Globally, the paper looks like a gathering of diverse sections written independently, leading to a non-uniform written and graphic quality.

Ultimately, I was left uncertain of what to make of the results. This is perhaps a useful outcome for a future research paper, but several sections are either unclear, incomplete, or presenting correction methods that were already published and tested (i.e., the oxygen correction method) while it is stated that "alternative correction methods" are used in this paper. That said, the statements in the "recommendations" section are worth making to the community, so I hope the authors resubmit this paper after a deep re-work of it.

*\*\* Minor issues \*\**

*Abstract*

L. 22: "… subject to varying .., and to biofouling".

L. 24: several is a quantifier and maintenance is an uncountable noun, they do not fit together. Please replace either the first (e.g., some) or the second (e.g., repairs) and modify the verb accordingly.

L.27: Please put in situ (and all other Latin words as e.g. for example) in italic

L.33: I suggest rephrasing "for future experiments, a more frequent sample collection would improve the data qualification and validation"

L.34: I suggest removing this sentence as it is more or less self-evident that data are provided at the end of each paper

*Introduction*

L. 41. ". Among other improvements, fixed ocean stations and SOOP were equipped with.."

L. 45: "… larger scales, because of the very sparse .."

L. 47: Not needed to put an s to ASV as Vehicles is already in its plural form

L. 56: "9-month-long"

L. 56: "… two wind-driven Saildrone ASV (SD) manufactured by Saildrone, Inc. (Alameda, CA) were used to …"

L. 59: What do the "problems" refer to? Please develop

L. 61: "long-duration"

L. 65: On what basis is this "different marine ecosystems" statement assigned?

L. 67: How do you define "high-quality data" here? What is the reference?

L.69: What kind of processes? Are all the processes studied hereafter? I suggest here to develop and/or focus this sentence by adding a link to the results/discussion section

*General comment on this Introduction section*: References and citations are missing. Among other papers, the authors could cite, for example, Delauney et al., 2010 (10.5194/os-6-503-2010) for the biofouling effect, Tanhua et al., 2019 for the "observational gaps" (10.3389/fmars.2019.00471), Gentemann et al., 2020 for the SailDrone description (10.1175/BAMS-D-19-0015.1) or Goni et al., 2010 (10.5270/OceanObs09.cwp.35) / Lüger et al., 2004 (10.1029/2003GB002200) for the SOOP program / $pCO_2$ sensor implementation.

*Experiment and data*

Section 2: I suggest replacing with "Material" and dividing the text into two sections:
2.1 "Data collection and experiment" with a brief description of the Atlan2med experiment and of the Saildrones. I believe that this could fit into 2 paragraphs, especially as the demonstration experiment has been already published.

2.2. "Comparative datasets". For each fixed station, several research papers have been already published. I suggest deeply shortening this section by adding references to published papers ("A more detailed description of the observation site can be found in…"). Except if the details of the station designs, sampling strategies, and analysis sequences/timing are not identical to the previously published papers, they do not need to be repeated here. I also suggest splitting this "comparative datasets" section rather by using the locations than the names, as proposed in Figure 1 (e.g., Liguro-Provencal basin facilities, North Adriatic region comparison sites…). Then, I would merge the glider paragraph with the LION and DYFAMED / W1M3A and E2M3A fixed stations parts into a Liguro-Provencal basin / North Adriatic basin sub-sections. This would reduce the text and clarify the locations of the sites and tools (especially because the glider transects are already cited in Section 2.2.).

L. 73: "… among numerous European academic institutions. A detailed description of the Atl2Med demonstration experiment can be found in Skejelvan et al. (2021)". I would reduce this paragraph by not listing all the institutes involved in the experiment.

L. 73: What are the exact deployment dates for the SDs? Please add this information

L. 83: What kind of characteristics? Please develop

L. 86: I suggest starting this paragraph with the main aims of the mission and then rephrasing: "The aim of the Atl2Med mission was to (1) study eddies in the Canary Current upwelling system off West Africa jointly with a vessel-based research expedition (RV Meteor M160) and (2) to validate the $CO_2$ measurements acquired at 6 fixed ocean stations (CVOO, DYFAMED, W1M3A, E2M3A, Miramare, and Paloma). This monitoring experiment was achieved with sensors and instruments installed on the SDs, but also equipment deployed at a number of facilities that were used to correct data from the SDs (see Section 3). Table 1… "

L. 91: ".. Table 2 indicates when SD maintenance was performed". I also suggest removing this table that is not needed as it could be summarized in 2 sentences.

L. 92: I suggest moving those two tables to the Supplementary Material (not crucial for the general understanding of the paper) "A detailed description of the instruments and sensors installed on the different features as well as their characteristics can be found in Tables 1 and 2 of the Supplementary Material".

L. 93: "A detailed description of the Atlan2Med .." (if the sentence is kept...)

L. 94. The citation should come earlier in the discussion, almost at the beginning.

L.98: Delete the "provided by Saildrone Inc." that is already said in the Introduction

L. 98: Explain the CTD acronym

L.99: ". This study focuses primarily on sensors acquiring temperature, salinity, dissolved oxygen, and $p$CO$_2$ data."

L. 104: Please add a reference for the ASVCO2 system.

L. 109: Is the measurement frequency only for $x$CO$_2$ measurements or for all the parameters? Please add precision.

Section 2.2: The LION fixed station is not cited in the first Section 2 paragraph' and it is unclear to me when those data have been used.

L. 131. "The main buoy"... That is?

L. 133-136: All those information about the sensors are given in Table 3. Also, the mooring line data are not used in the paper so the sentence about it can be removed.

Table 5: I suggest moving Table 5 to the Supplementary Material

L. 143: The paragraph could be shortened and cut after the "… online server." sentence. All the other information are either given in the Table or not needed as papers about the infrastructure and the network functioning have been already published.

L. 157: Please add a reference for the HydroC system

L. 167: ".. transmit them".

L.170, 182 & 189: Same as at lines 133-136

Section 2.6: I suggest modifying the title to "shipboard data" or similar. Also, please add the name of the campaign in this section (Meteor M160 cruise if I am right)

L. 194: Delete "furthermore"

L. 196: "Table 5 gives an..". In the Table, the M160 cruise (or R/V Meteor) is not indicated. I suppose that it is what the Geomar name in the facility column refers to, but I suggest being more precise

L. 214: In the legend of the Table, DIssolved → Dissolved

L. 226: "The Copernicus … et al., 2021) products. Daily data were..."

*Methods*

L. 235: "… intervention, salinity, dissolved oxygen, and Chl-a values .."

L. 231-233: Please add references to support this statement

L. 239: Please add a comma between data and we focus

L. 240: $p$CO$_2$ data are not listed in the above paragraph as incorrect data... Conversely, Chl-a data are cited but do not seem to be included in the correction methods presented after. Please clarify

L.241-242 & Table 6: This description of the temperature sensor behavior should go earlier, before the list of all the parameters that need to be adjusted (line 235 I would say). What would be the explanations for these apparent temperature and salinity accuracy differences as the same sensor is measuring those parameters?
→ Worth pointing out that this 7-crossover comparison might not reflect the entire sensor consistency over its entire deployment time and is not statistically sufficient. A reference to the time-series plot (Fig. 4a) with the two SDs temperature records would be useful to highlight the stability and consistency of the signal. Here again, I suggest putting Table 6 in the Supplementary Material.

L.242 glides → gliders

L. 250: A space is missing between "… .1b), the SD showed..."

L. 249 & Figure 2: A superposition of the two SDs salinity time-series would support this statement

L. 256: "… and T5, salinity shifts of 1 were observed…"

L. 261: add a comma after correlation. How was defined this time/distance criterion?

L.263: What was the criterion to define the "nearest node"?

L. 265: ".. large time span since the last maintenance". I do not agree with this statement that is incorrect for the SD 1053 considering the transects 3, 4, and 5, particularly (Fig. 2b).

L. 266: on the one hand… on the other hand... The comparison is incomplete, please modify or replace the adverb

L. 269: have shown

L. 274: The Gibraltar Strait

L. 276: "… variability. SD 1053 also showed considerable…" → What would be the explanation/hypothesis behind this observation?

L. 280: Remove the space between the model and the comma

L. 283: show

L. 284: "over the 60 days of measurements."

L. 289: ".. was of 0.26.". Also, here you give 2 digits for salinity values while there is only 1 digit given (and on Fig. 3). Please modify and harmonize.

L. 290: When? Which transect are you talking about?

L. 291: I am confused here, only Figs. 3i and j are cited but the transects 2 to 5 are listed…

L. 292: Mean differences? "..differences of" without the comma
It's confusing here as it is unclear if you are talking about the (mean?) differences or the offsets derived from the linear regressions…

L. 299: Where is the comparison with the fixed ocean stations?

Sections 3.1 & 3.2 are my biggest concern about the methods section. This regression analysis comes off as very sloppy. No statistics are given, only $R^2$ values. And for the $R^2$ values no information is given about the hypothesis tested and the test used... Trends are sometimes presented or cited but without the trend's values and/or uncertainties. There is no mention of how the trends are derived in Section 3.2 (though one would assume an ordinary least squares regression as in Section 3.1, which I'm not entirely sure is appropriate here (neither in Section 3.1)). Looking at the data distribution over time in Figure 3, I question how robust the regressions are so I'd like to see uncertainties for the regression line visualized. See for example Fransner et al (doi:10.5194/bg-2020-339) for an example of how to do this. There is some discussion around these results, but again quite superficial and that is for all the parameters cited/corrected. On lines 362-363, it is mentioned that similar periods of over and under saturation can be observed, but it is unclear what the relevance of that is and also where those saturation values were measured. This analysis needs considerably more care and interpretation before publication.

L. 298: I suggest adding a methodology section or at least some explanations about the method used, the tests done, and so on. Also, could you please indicate (maybe on the plots or in an additional table) the final offset and drift values obtained (+ the statistics)

L. 298 & 504: It seems that the US-English is used in the manuscript elsewhere → replace z by s

L. 302: Add a comma after temperature

L. 304: Add a reference here

L. 307: Something eastern is written with a capital letter, sometimes no. Please harmonize

L. 310: "… values. This procedure…"

L. 311: The reference Takeshita et al., 2013 is not in the reference list

L. 313: Please harmonize the units over the entire manuscript, especially for the oxygen. In Table 4, the sensor accuracy is given in µmol/kg (molinity) while in Section 3.2 the µmol/L (molarity) unit is used. The International System of Units in Oceanography (ISO) report published by Unesco (1985) recommended using moles per kilogram of solution for dissolved gases: "*For concentrations of dissolved gases units such as milliliter or cubic centimeter per liter at the reference temperature and pressure should also be discarded. Such concentrations should henceforth be reported uniformly in moles per cubic decimetre, or in moles per kilogram of solution.* (p.120)"

L. 315: "… 1053, respectively." Give the exact $R^2$ value

L. 315: Give the exact $R^2$ value. What is the statistical test used, and between which variables? An $R^2$ of 0.6 is quite high, is it not? Sixty percent of the variance in oxygen can be explained by temperature variations, thus, if I understand well, I do not agree with the statement "independent of temperature". I also feel this sentence is written oddly (I don't get really what the temperatures refer to). I suggest splitting this sentence into two. Fig. 4e and f refer to oxygen concentrations, not oxygen saturations…

L. 317: there is a space between °C and the comma

L. 322: What is the criterion defining "unreasonable" data?

L. 325: The authors stated (L. 322) that "no significant trend in sensors response" (according to Figures g & h it seems true) is observed (once again, what is the statistical test used here?) and then claimed that they want to "correct the negative trend". Please clarify and be consistent

L. 335: In the original vapor pressure of water equation (Johnson et al., 2015), the natural logarithm (ln) term is used while in the manuscript the logarithm (log) is written in the equation. Please check the equation and the calculations done.

L. 336: What is the unit for the volume fraction of oxygen, what is the value used? If it is a constant value (I believe that it might be $20.946 \pm 0.002$ percent), then the authors should write it

L.335, 338, 340 & 343: I suggest numbering the equations. Also, what are the units?

L. 339: G is the gain factor/gain correction.

L. 339: "The Epp was..."

L. 341: The term SDcsd does not correspond to the what is written in the equation. I suggest clarifying this sentence here as follows "The corrected oxygen concentration ($O_2$csd) from the SDs was calculated...". Data is a plural noun → "are/were" (and elsewhere in the manuscript). When SDs oxygen data not corrected are used, I suggest specifying "raw data" ("… from adjusting raw oxygen data measured by the SDs..").

L. 344: Here again, the sentence is very confusing and does not correspond to what was written previously. It is clearly stated in L. 316 that there is no temperature dependency... From my understanding, the purpose of the correction is not to detrend a time-series (that would imply a consistent drift/change over time, which is not the case here (Fig 4 g-h)) but to correct oxygen sensor instabilities and drifting over time as the gain factor is updated to correct for temporal drift.

L. 346: How these dissolved oxygen saturations deterministic (I assume) trends were removed?

L. 348-353: Here again, it is unclear how dissolved oxygen data were detrended, how the relationships were calculated, what the statistical relevance of the calculations was... The last correction step (i.e. the biological activity effect removal) relying on the assumption that only biological processes drive the $O_2$ content variability seems to induce an over-estimation of these processes as air-sea $O_2$ fluxes and physical structures (convection, eddies) could significantly impact the $O_2$ reservoir. A more detailed discussion (is it in agreement with the literature?) about this assumption (and its associated error) would be at least a way forward to add confidence. I am also unsure about the data used here... I would use the corrected oxygen data and then derive the residual values, as I have the feeling that otherwise an overestimation of the biological activity could occur.

Section 3.2: The conversion between equilibrium partial pressure, $pO_2$, and seawater $O_2$ concentration, depends on the seawater salinity (see Bittig et al., 2016 & 2018). Was this salinity correction done?

L. 360: Remove "the" Spring 2020.

L. 361: Why is it "particularly interesting"?

L. 366: I would remove the term "sensor" before "$pCO_2$ measurements" as it could lead to confusion with the SD $pCO_2$ system and suggest another term like "fixed-sites $pCO_2$ measurements" (feel free to ignore this comment, this is a matter of personal perception). At least please harmonize the nomenclature over the entire paragraph (line 380 they are called "stations sensors")

L. 368: "... Atlan2Med cruise/campaign/experiment"

L. 370: Please develop this "minor variability"

L. 376: Depending on the input variables/combination used to derive the $pCO_2$ (as well as on the equilibrium constants), the uncertainty could vary a lot, from $\pm$ 1.8 µatm to $\pm$ ~6 µatm (e.g., Millero, 1995, Orr et al., 2018). Please justify your choices.

L.377: Please develop the "the hydrogen fluoride constant KF" and put the F as a subscript

L. 378: Please put the letter "p" in italics (and elsewhere in the manuscript)

L. 394: "... (in µatm)"

L. 397: Considering Figure 6b, is it still accurate to consider an associate uncertainty of $\pm$ 5 µatm?

Section 3.3 and 3.4: I would suggest putting them as sub-sections (3.3.1 & 3.3.2) as a bigger one merging them and entitled, for example, "Correction and adjustment of $p$CO$_2$ data". I would thus rename Section 3.3.1. to be more precise "Fixed-sites $p$CO$_2$ data acquisition and qualification".

*Results and discussion*

Section 4.1: For salinity data, the correction is discussed in this "results and discussions" section while for the other parameters (oxygen and $p$CO$_2$), corrected data are already presented and discussed in the Methods section. Please harmonize the data presentation.

L. 404: Add a comma after "(T1)"

L. 407: "The salinity correction induced an over-estimation /over-estimated the salinity values. Instead, raw salinity data were in …"

L. 411: Please add a reference to the plot you are referring to.

L. 411: "with respect to"

L. 414: "… (Fig. 7e). The corrected salinity …"

L. 418: "… between the E2M3A fixed ocean station ...". In this sentence, you are comparing a structure itself to data. Please be clear: "fixed ocean station dataset and glider measurements"

L. 420: "poor": I suggest removing those kinds of subjective adjectives that are uncountable and sound not scientific. Replace by "non-significant" and present a strong argument (statistics)

L. 422: This plot does not show trends! Please use the words temporal changes, variations, (similar) dynamics

L. 423: "… June 1$^{st}$… April 27$^{th}$ …"

L. 426: "… are consistent… ; differences being mainly due to the distance …"

L. 428: "Considering that during T1…to apply the salinity correction after this transect."

L. 432: "… discrete dissolved oxygen measurements were not available … Thus, the SD corrected …"

L. 435: Please name the two locations chosen

L. 436: L. 235, the authors claimed that Chl-a values showed inconsistencies, while here they are used. It leads to confusion: I suggest moving L.460 upward
Moreover, this analysis comes off as very simplistic as it relies only on a visual inspection

L. 439: Please add a reference to support this statement

L. 440: Here again, please add concrete data to support this "strong linear correlation" statement

L. 441: What does this "slightly lower/higher" represent? Please add a reference

L. 444: Is it always the case? Even during intense wind episodes? (Ulses et al., 2020)

L. 446: Are the authors describing here the oxygen saturation in the Mediterranean Sea? I suggest applying this example/sentence to Mediterranean Sea data

L. 449: How is the evaluation done?

L. 451-452: The DOI can be removed from the main text and put in the data availability section

L. 456: Same remark here for the dates than at Line 423

L. 457: were high

L. 458: "… lines). The optical sensors on the SDs…"

L. 461: "… needs weeks .."

L. 462. "We refer to…". Please remove the thus

L. 464: Please explain the link between Chl-a data and the temperature, or at least modify the sentence as you are starting with "the patch of high Chl-a" and continuing with "evident in sea surface temperature".

L. 482: The decrease in oxygen is not observed for both SDs, only on the SD 1053 time-series.

L. 488: A reference would be appreciated here to support this statement

L. 494: A brief discussion about this relationship, varying regionally, and its consequences (i.e., the T sensitivity is larger in colder regions and lower in the warmer tropics) would maybe be interesting here (see Gallego et al., 2018)

L. 500: What were the input variables at the other sites? What would be the uncertainty associated with the other difference estimates?

L. 504: Would it be possible to estimate the impact of those processes?

*Data availability*

Section 5 & Table 8: I suggest moving this section and the table to the Supplementary Material. In Table 8, please split the DOI column into two columns, one only with the DOI and the other with the references.

*Summary*

L. 519: "… covering both the Eastern Tropical North Atlantic region …". Also, as the ETNA region is cited numerous times, maybe the acronym could be used here and previously

L. 529: sensors on the SDs (also L. 531, 556, 562). Moreover, what is corrected is not the sensor itself but the data produced by it. Please correct

L. 531: I do not agree with this sentence: the oxygen data correction method is not new (Johnson et al., 2015, Bittig et al., 2018)

L. 533: "… data sets that are available…". Please harmonise the way the word dataset is written over the entire manuscript (sometimes with a space, sometimes no)

L. 535: the paragraph begins with a reading out of the limitations, but it seems that there is only one issue considering the rest of the paragraph. I suggest rephrasing or modifying the first word

L. 536: "… cruises, etc."

L. 538: "Consistent" is an overstatement as no statistics are associated with the results. I also suggest rephrasing the sentence as it is more or less written two times that the consistency is fine…

*Future and recommendations*

L. 549: I would move the beginning of this Section from line 544 to 549 in the conclusion section, reducing its content and rephrasing some parts of the conclusion to fit in. Indeed, no recommendations or pieces of advice are given here, only a repetition of the previous statements

L. 556: The RBR acronym is not defined in the manuscript

L. 565: Please rephrase... "the sensor mounting should ensure that the sensors are mounted" sounds repetitive

L. 566: "… from a lack of …"

For this section, I would suggest summarising the recommendations via a bullet list or at least by numbering them. It would clarify what the improvements from this experiment are based on your observations, which is unclear in the current Section 7

*\*\* Figures & Tables \*\**

Figure 1: The bottom panel seems to have been warped. Please add on this one some indications about the oceans/seas, countries, and the lat/long information for the x/y axis. For the upper panel, the legend is incomplete: Argo float symbols and the two SDs colors are detailed, but the symbols for the fixed-stations, the glider, and the research vessel are not explained. I suggest adding on the small maps the names of the fixed sites and small dashed lines to link the "zoom windows" to the small squares on the map. I also suggest shifting them a little bit to better show the main map.
The two blue lines (T1 & T2) are superimposed, but I thought the two SDs were deployed at the same time… Also, the transect cutting is incomplete: to what are the data (SD 1030) between December and January qualified/associated with?

Table 2: Not needed, could be completely removed and replaced by 2 sentences in the main manuscript. Also, the backslash (\) is not in the same direction as the others in the manuscript (column 1 first box)

Table 3: In the main text, SBE16 plus is written with a +. Please standardize. Make uniform the units too, with either a -1 as a superscript or a /

Figure 2: Would be useful to add on the Figure the transects (T1, T2, …). Please modify the legend ".. and Argo float data.." . The legend on the figure (i.e. stations) is confusing as it could refer to the fixed station sites described before. Please clarify. I also suggest adding a sentence in the main manuscript stating that in the manuscript salinity values are expressed in PSU (then remove this unit in the figures).

Figure 3: The y-axis is not labeled ($\Delta S$?). Also, I suggest harmonizing the y-axis range/scale to avoid confusion. What is the statistical test used? See my previous comment about that. Colors for each SD are not the same between Figure 2 and Figure 3 (blue and purple against black and red): please harmonize. A map on the side would be relevant to better visualize where the data were acquired.

Figure 4: It seems that the red color is used for the SD 1053 and the black one for the SD1030, but in panels g and h it is the reverse. Please harmonize. I suggest reducing this Figure by (1) putting the temperature panel with Figure 2 (as temperature data are cited earlier in the paper at the beginning of Section 3, (2) removing the panels e and f (not necessary, could be put in the Supplementary Material) and (3) moving upward the panels e and f as this section is first and foremost dedicated to oxygen data. I also suggest adding the statistical results to these panels. In Figure 4b, please reduce the thickness of the black trend line as the grey one is currently not visible. The y axes are not completely labeled (Oxygen Saturation (%), $O_2$ (µmol/l)). What are the dashed grey lines?

Figure 5: The y axes are not completely labeled (Oxygen Saturation (%), $O_2$ (µmol/l), temperature...). In Figure D, the legend box is on the x-line. I suggest reducing a lot this figure by removing some panels (e.g. the detrended and not detrended, especially as it is unclear in the main manuscript...). Once again here, I suggest moving the "corrected $O_2$ data" time-series plot to the top as this is what this section is about. Please harmonize the legend in the boxes (correction/corrected...). In the legend, please correct "back → black"

Figure 6: I suggest completing the y-axis for panel b by adding $\Delta$ (i.e. $\Delta p\text{CO}_2$ (µatm)) to avoid any confusion with panel A.

Figure7: Colors are almost indistinguishable on the upper panel (Fig. 7a). I also do not think that it is relevant here to plot the model salinity data as SD data were corrected using them... They obviously more or less match them. I suggest not presenting the data with multiple panels as it is done now (corresponding to certain transects) but rather using the current panels A and B and adding the other reference measurements on them. If they are bigger the dots will be big enough and the main message clear. Thus, it would delete the panels c to g. Please also use another for the floats and the glider data (both of them are yellow…). Following the text structure, I would first present the data recorded by the SD 1053 (first discussed) and then by the SD 1030. Numbers on the panels (corresponding to the distances) are unreadable. If the authors want to keep the panels as they are currently, I suggest at least using the same y-axis scale (Fig. 7 c-e).

Figure 8: Units are missing. The y-axes as well as the colorbar are incomplete. The legend is incomplete too as the second y-axis in Figure 8b is not explained.

Figure 9: One panel is enough for both the sea surface temperature and the vertical section. It is not needed to put the three dates as the plots are similar. Here too, units are missing. The y-axes as well as the colorbar are incomplete. The addition of the SDs pathways on panel C would be useful to understand the differences in $O_2$ concentration observed between the two platforms.

Figure 10: Are $fCO_2$ data used or $pCO_2$? The legend is not in agreement with the plots and Section 4.3. In the boxes, please write $pCO_2$ rather than $CO_2$. In Figure 10B, why dots (that are indistinguishable) are not linked?

Table 8: A "E" is missing line 6 column 3 "MOOS program". I suggest renaming the first column "Platform" or similar

Generally, the number of plots (that is too high!) for each parameter studied is not well balanced. Even if it does not have to be equal, there is a strong disproportion among the figures, with for example 4 figures for the salinity, 15 for the oxygen, and only 2 for the $pCO_2$ in the methods section.

General remark: Please have a look at the colors used for all figures and make sure they are colorblind-friendly (I do believe that it is not the case for Figures 8 & 9).

---

## Author Response (AR1)

Dear Reviewer,

Sorry for not providing a track change version of the manuscript. The manuscript was completely revised, and we thought we didn't have to do it.

We thank you for assessing our manuscript and for all the time and effort dedicated to it, we have integrated all of your proposed corrections into the text.

Please find below our reply comments, including a list of the changes made in the manuscript, and our revised manuscript, with modified text highlighted in orange.

Kind regards,

Riccardo Martellucci on behalf of the authors

*** General ***

*The paper by Martellucci et al. presents a study of the correction applied to several biogeochemical parameters acquired by two SailDrone platforms during the Atlan2Med demonstration experiment. The two platforms sailed from the Eastern Tropical Atlantic Region to the North-Western Mediterranean Sea between July 2019 and October 2020. The authors have put together an interesting new dataset and reviewed choices regarding how and when data adjustments were done. Their findings show significant offsets between raw data and reference datasets from nearby cruises and stations. They arrive at a plausible set of conclusions and suggest useful corrections. The resulting paper could be a useful contribution to the literature as it is needed for deeper research, but I believe it should be returned to the authors for deep revisions and changes, as explained below.*

*** Major comments ***

*1. The manuscript has been submitted as an original research paper, but its content is rather technical and descriptive, and the discussion is quantitative rather than qualitative, and did "feel" technical. I recommend submitting the paper as a technical note, even if in its actual form, this is too long to be accepted in this format. The paper should focus solely on detailing the data qualification and correction, omitting particularly the site descriptions (already published), several figures, and tables that could be either completely removed or put to the supplementary material. This paper needs to be shortened and focused. In its current form (without real comparison or added-value), the paper does not meet the publication criteria.*

We have restructured and shortened the manuscript as suggested by the referee. This is particularly visible for section 2 Material, where some of the material in moved to the Supplementary Material, some of the material is replaced by references, and some of the material is merged. We have also added a dataset from RV Ucadiz (Spanish research vessel), which was not included in the original manuscript version.

The restructure also includes moving some of the text from the Methods section to the section on Results and discussion, where it belongs. This introduced more balance between salinity, dissolved oxygen and pCO2 in the manuscript.

We have reorganized the sections Summary and Experiences and recommendations with aim to avoid repetition and to clarify the messages we want to highlight.

*2. My main concern in this paper is the corrections presented in Section 3 and the lack of robust statistics. It is not clear what hypothesis has been tested, nor which test is used. The authors need to include several other statistics (such as p-value, and RMSE) for their regressions. There is also no detail on how the regressions were determined (ordinary least squares I assume, but why is that appropriate for some parameters such as the temperature that has so few data points?). In Section 3.2 the authors claim several times that oxygen data have either a temperature dependence or independence and seem to construct the entire correction based on this observation (not statistically proven). It really does not look like that in Figure 4. This claim needs to be substantiated, and the methods used to determine this evaluation detailed. Based on the very little information given on methods and assumptions I do not believe these results are valid... and that leads me to question the analysis done using Argo data.*

The Methods section was improved and clarified according to suggestions from the referee. The robustness of the statistics is severely improved, and we have inserted a table showing result of the salinity statistics performed, including significance level, distribution, correlation coefficient, and root mean square error.

*3. The quantitative aspects seemed, in places, potentially incorrect (i.e., the temperature and salinity data analysis). Comparisons with the literature are missing in a lot of places, leading to a confusing feeling about the accuracy of the corrected datasets. In other places, comparisons against data are done while it was stated previously that the data were either incorrect (Chl-a data) or acquired at a distance too far from the SD track (float data). A comparison with historical datasets would be a way forward to reduce this feeling of an incomplete and somehow superficial analysis. Generally, several references are missing, in both the discussion section and in the introduction to better describe the studied area and its characteristics.*

We have inserted more references throughout the manuscript, and we have also harmonized the units all over. We compared our dataset with climatological dataset.

*4. Some of the figures are missing information, some others could be split and simplified, and the writing is difficult to understand in some places (but excellent in other parts). Some of the notation is inconsistent (see line-by-line comments). Globally, the paper looks like a gathering of diverse sections written independently, leading to a non-uniform written and graphic quality.*

We have improved the figures as suggested by the referee and ensured that there is a balance between the salinity, dissolved oxygen and pCO2 figures in the manuscript.

*Ultimately, I was left uncertain of what to make of the results. This is perhaps a useful outcome for a future research paper, but several sections are either unclear, incomplete, or presenting correction methods that were already published and tested (i.e., the oxygen correction method) while it is stated that "alternative correction methods" are used in this paper. That said, the statements in the "recommendations" section are worth making to the community, so I hope the authors resubmit this paper after a deep re-work of it.*

We hope that this revised version is in agreement with your suggestions and suitable for publication in Earth System Science Data.

*** Minor issues ***
*** Abstract***
*L. 22: "... subject to varying .., and to biofouling".*

We changed the sentence as follows:

The sensors on board were exposed to varying degrees of degradation and biofouling depending on the geographical area and season, which led to a deterioration of the measurements.

*L. 24: several is a quantifier and maintenance is an uncountable noun, they do not fit together. Please replace either the first (e.g., some) or the second (e.g., repairs) and modify the verb accordingly.*

Thanks, we changed "several" with "some".

*L.27: Please put in situ (and all other Latin words as e.g. for example) in italic*

Thanks for the suggestion, we wrote all the "*in situ*" in italic.

*L.33: I suggest rephrasing "for future experiments, a more frequent sample collection would improve the data qualification and validation"*

We replaced the sentence as suggested.

*L.34: I suggest removing this sentence as it is more or less self-evident that data are provided at the end of each paper*

We removed it.

*Introduction*

*L. 41. ". Among other improvements, fixed ocean stations and SOOP were equipped with.."*

We replaced the sentence as suggested.

*L. 45: "... larger scales, because of the very sparse .."*

We replaced the sentence as suggested.

*L. 47: Not needed to put an s to ASV as Vehicles is already in its plural form*

Thanks, we replaced it in the text.

*L. 56: "9-month-long"*

We replaced it.

*L. 56: "... two wind-driven Saildrone ASV (SD) manufactured by Saildrone, Inc. (Alameda, CA) were used to ..."*

We replaced the sentence as suggested.

*L. 59: What do the "problems" refer to? Please develop*

We add this sentence:

SDs are prone to errors primarily due to sensor drift, which can be caused by either biofouling or malfunctioning sensor parts.

*L. 61: "long-duration"*

We replaced it.

*L. 65: On what basis is this "different marine ecosystems" statement assigned?*

We change the sentence as follows:

Furthermore, the demonstration experiment allowed us to focus SD measurements on different marine environments, the Atlantic Ocean and the Mediterranean Sea

*L. 67: How do you define "high-quality data" here? What is the reference?*

We change the sentence as follows:

The experiment additionally evaluated the ability of such ASV to provide data with sufficient quality to be relevant for the scientific community.

*L.69: What kind of processes? Are all the processes studied hereafter? I suggest here to develop and/or focus this sentence by adding a link to the results/discussion section*

Thanks for the suggestion, we rewrote the sentence as follows:

The objective of the present work is to evaluate and correct the data collected by the SDs in order to provide a homogenised and comparable data set useful for the study of processes such as air-sea gas exchange in the Atlantic Ocean and Mediterranean Sea. While this paper focuses on the methods, a follow up paper will focus on biogeochemical processes occurring in the area.

*General comment on this Introduction section: References and citations are missing. Among other papers, the authors could cite, for example, Delauney et al., 2010 (10.5194/os-6-503-2010) for the biofouling effect, Tanhua et al., 2019 for the "observational gaps" (10.3389/fmars.2019.00471), Gentemann et al., 2020 for the SailDrone description (10.1175/BAMS-D-19-0015.1) or Goni et al., 2010 (10.5270/OceanObs09.cwp.35) / Lüger et al., 2004 (10.1029/2003GB002200) for the SOOP program / pCO2 sensor implementation.*

We added the reference as suggested.

*Experiment and data*

*Section 2: I suggest replacing with "Material" and dividing the text into two sections:*
*2.1 "Data collection and experiment" with a brief description of the Atlan2med experiment and of the Saildrones. I believe that this could fit into 2 paragraphs, especially as the demonstration experiment has been already published.*

We rebuilt the section as suggested.

*2.2. "Comparative datasets". For each fixed station, several research papers have been already published. I suggest deeply shortening this section by adding references to published papers ("A more detailed description of the observation site can be found in…"). Except if the details of the station designs, sampling strategies, and analysis sequences/timing are not identical to the previously published papers, they do not need to be repeated here. I also suggest splitting this "comparative datasets" section rather by using the locations than the names, as proposed in Figure 1 (e.g., Liguro-Provencal basin facilities, North Adriatic region comparison sites…). Then, I would merge the glider paragraph with the LION and DYFAMED / W1M3A and E2M3A fixed stations parts into a Liguro-Provencal basin / North Adriatic basin sub-sections. This would reduce the text and clarify the locations of the sites and tools (especially because the glider transects are already cited in Section 2.2.).*

We shortened and merged the paragraph inserting references, as suggested.

*L. 73: "… among numerous European academic institutions. A detailed description of the Atl2Med demonstration experiment can be found in Skejelvan et al. (2021)". I would reduce this paragraph by not listing all the institutes involved in the experiment.*

We did that.

*L. 73: What are the exact deployment dates for the SDs? Please add this information*

We specify the time interval:

place between 18 October 2019 and 17 July 2020

*L. 83: What kind of characteristics? Please develop*

We change the sentence as follows:

During the experiment, the SDs crossed the ETNA region, the Strait of Gibraltar, and the northern part of the western and central Mediterranean Sea including the Ligurian Sea, the Strait of Sicily, the Strait of Otranto, and the Adriatic Sea (Fig. 1).

*L. 86: I suggest starting this paragraph with the main aims of the mission and then rephrasing: "The aim of the Atl2Med mission was to (1) study eddies in the Canary Current upwelling system off West Africa jointly with a vessel-based research expedition (RV Meteor M160) and (2) to validate the CO2 measurements acquired at 6 fixed ocean stations (CVOO, DYFAMED, W1M3A, E2M3A, Miramare, and Paloma). This monitoring experiment was achieved with sensors and instruments installed on the SDs, but also equipment deployed at a number of facilities that were used to correct data from the SDs (see Section 3). Table 1 ... "*

Thanks for this suggestion, we integrated in the text.

*L. 91: ".. Table 2 indicates when SD maintenance was performed". I also suggest removing this table that is not needed as it could be summarized in 2 sentences.*

We moved the table to the supplementary material.

*L. 92: I suggest moving those two tables to the Supplementary Material (not crucial for the general understanding of the paper) "A detailed description of the instruments and sensors installed on the different features as well as their characteristics can be found in Tables 1 and 2 of the Supplementary Material".*

Thanks for the suggestions, we did that.

*L. 93: "A detailed description of the Atlan2Med .." (if the sentence is kept...)*

We deleted the sentence.

*L. 94. The citation should come earlier in the discussion, almost at the beginning.*

Thanks, we followed this suggestion.

*L.98: Delete the "provided by Saildrone Inc." that is already said in the Introduction*

We deleted it.

*L. 98: Explain the CTD acronym*

We did that.

*L.99: ". This study focuses primarily on sensors acquiring temperature, salinity, dissolved oxygen, and pCO2 data."*

Thanks for the suggestion, we added this sentence.

*L. 104: Please add a reference for the ASVCO2 system.*

We did that.

*L. 109: Is the measurement frequency only for xCO2 measurements or for all the parameters? Please add precision.*

This information is inserted in Table S3 in the Supplementary material.

*Section 2.2: The LION fixed station is not cited in the first Section 2 paragraph' and it is unclear to me when those data have been used.*

Sorry for this, we removed the LION fixed station as it was not used.

*L. 131. "The main buoy"... That is?*

This is the W1M3A.

*L. 133-136: All those information about the sensors are given in Table 3. Also, the mooring line data are not used in the paper so the sentence about it can be removed.*

We removed the sentence.

*Table 5: I suggest moving Table 5 to the Supplementary Material*

We did that.

*L. 143: The paragraph could be shortened and cut after the "… online server." sentence. All the other information are either given in the Table or not needed as papers about the infrastructure and the network functioning have been already published.*

Thanks for the suggestion, we rebuilt the paragraphs.

*L. 157: Please add a reference for the HydroC system*

We did that in Table S2

*L. 167: ".. transmit them".*

We removed this.

*L.170, 182 & 189: Same as at lines 133-136*

We did it.

*Section 2.6: I suggest modifying the title to "shipboard data" or similar. Also, please add the name of the campaign in this section (Meteor M160 cruise if I am right)*

Thanks for the suggestion, we did it.

*L. 194: Delete "furthermore"*

We removed this.

*L. 196: "Table 5 gives an..". In the Table, the M160 cruise (or R/V Meteor) is not indicated. I suppose that it is what the Geomar name in the facility column refers to, but I suggest being more precise*

The Meteor cruise number (M160) is included in the text, and Table 5 is moved to the Supplementary Material (Table S4). This table refers to analysing methods for DIC and TA, and GEOMAR is included here since the samples were analysed on shore after the cruise.

*L. 214: In the legend of the Table, DIssolved → Dissolved*

Thanks, we corrected it.

*L. 226: "The Copernicus … et al., 2021) products. Daily data were…"*

Thanks for the suggestion, we replaced the sentence as suggested.

*Methods*

We rebuilt the methods section in three sub-paragraphs. In the main text I colored the sub paragraph headings in orange. Many corrections have been made and the text has changed from the previous one.

Regarding the salinity correction, we first compared the data with climatological dataset, and then we compare the SD data with the model output.

In case there was no response to the comment is because it was moved or deleted

*L. 235: "… intervention, salinity, dissolved oxygen, and Chl-a values .."*

This sentence is removed.

*L. 231-233: Please add references to support this statement*

We removed this sentence, however we insert specific citations for every geographic location we discussed in the text.

*L. 239: Please add a comma between data and we focus*

This sentence is rewritten.

*L. 240: pCO2 data are not listed in the above paragraph as incorrect data... Conversely, Chl-a data are cited but do not seem to be included in the correction methods presented after. Please clarify*

Thanks for the suggestions, we added this sentence:

The optical sensors on the SDs and thus, the Chl-a measurements, were strongly affected by biofouling for most of the demonstration experiment, which is why we do not use these measurements in this work. However, during the 10 first days in October 2019, the Chl-a data acquired by the SDs seemed to produce reasonable values in accordance to Delory et al. (2018), who found that for new sensors the increase in biofouling needs weeks to become significant. We refer to these Chl-a data, collected by the SDs in the transect T1, when explaining the dissolved oxygen oversaturation episode off the Canary Islands.

*L.241-242 & Table 6: This description of the temperature sensor behavior should go earlier, before the list of all the parameters that need to be adjusted (line 235 I would say). What would be the explanations for these apparent temperature and salinity accuracy differences as the same sensor is measuring those parameters?*

*Worth pointing out that this 7-crossover comparison might not reflect the entire sensor consistency over its entire deployment time and is not statistically sufficient. A reference to the time-series plot (Fig. 4a) with the two SDs temperature records would be useful to highlight the stability and consistency of the signal. Here again, I suggest putting Table 6 in the Supplementary Material.*

Thanks for this, we have taken the suggestion into consideration and integrated it into the text and modified the figures.

*L.242 glides → gliders*

Thanks, we did that.

*L. 250: A space is missing between "... .1b), the SD showed..."*

The sentence is rewritten.

*L. 249 & Figure 2: A superposition of the two SDs salinity time-series would support this statement*

We did this.

*L. 256: "... and T5, salinity shifts of 1 were observed..."*

The sentence is rewritten.

*L. 261: add a comma after correlation. How was defined this time/distance criterion?*

The sentence is rewritten.

*L.263: What was the criterion to define the "nearest node"?*

we have changed the sentence to make it clearer:

The nearest nodes (in km) with respect to the model data grid to the SD trajectory were chosen.

*L. 265: ".. large time span since the last maintenance". I do not agree with this statement that is incorrect*
*for the SD 1053 considering the transects 3, 4, and 5, particularly (Fig. 2b).*

Thank you for the suggestion, the two vehicles were analyzed separately

*L. 266: on the one hand... on the other hand... The comparison is incomplete, please modify or replace*
*the adverb*

The sentence is rewritten.

*L. 269: have shown*

The sentence is rewritten.

*L. 274: The Gibraltar Strait*

Thanks for this, we corrected in the text.

*L. 276: "... variability. SD 1053 also showed considerable…" → What would be the explanation/hypothesis behind this observation?*

The sentence is rewritten.

*L. 280: Remove the space between the model and the comma*

The sentence is rewritten.

*L. 283: show*

The sentence is rewritten.

*L. 284: "over the 60 days of measurements."*

The sentence is rewritten.

*L. 289: ".. was of 0.26.". Also, here you give 2 digits for salinity values while there is only 1 digit given (and on Fig. 3). Please modify and harmonize.*

Thanks, this is harmonized.

*L. 290: When? Which transect are you talking about?*

The sentence is rewritten.

*L. 291: I am confused here, only Figs. 3i and j are cited but the transects 2 to 5 are listed… L. 292: Mean differences? "..differences of" without the comma*
*It's confusing here as it is unclear if you are talking about the (mean?) differences or the offsets derived from the linear regressions…*

The sentences are rewritten and clarified.

*L. 299: Where is the comparison with the fixed ocean stations?*

This is now clarified in the text.

*Sections 3.1 & 3.2 are my biggest concern about the methods section. This regression analysis comes off as very sloppy. No statistics are given, only $R^2$ values. And for the $R^2$ values no information is given about the hypothesis tested and the test used... Trends are sometimes presented or cited but without the trend's values and/or uncertainties. There is no mention of how the trends are derived in Section 3.2 (though one would assume an ordinary least squares regression as in Section 3.1, which I'm not entirely sure is appropriate here (neither in Section 3.1)). Looking at the data distribution over time in Figure 3, I question how robust the regressions are so I'd like to see uncertainties for the regression line visualized. See for example Fransner et al (doi:10.5194/bg-2020-339) for an example of how to do this. There is some discussion around these results, but again quite superficial and that is for all the parameters cited/corrected. On lines 362-363, it is mentioned that similar periods of over and under saturation can be observed, but it is unclear what the relevance of that is and also where those saturation values were measured. This analysis needs considerably more care and interpretation before publication.*

Following your suggestions, we rewrote the paragraphs.

*L. 298: I suggest adding a methodology section or at least some explanations about the method used, the tests done, and so on. Also, could you please indicate (maybe on the plots or in an additional table) the final offset and drift values obtained (+ the statistics)*

We added the information in Figure 3.

*L. 298 & 504: It seems that the US-English is used in the manuscript elsewhere → replace z by s*

We did it.

*L. 302: Add a comma after temperature*

This is done.

*L. 304: Add a reference here*

Thanks for the suggestion, we added this sentence:

During the demonstration experiment, sea temperature (Fig. 4a) showed a seasonal signal similar to those observed at these latitudes (Pastor et al., 2019).

*L. 307: Something eastern is written with a capital letter, sometimes no. Please harmonize*

We have harmonized this.

*L. 310: "… values. This procedure…"*

This sentence is rewritten.

*L. 311: The reference Takeshita et al., 2013 is not in the reference list*

We add this in the references section.

*L. 313: Please harmonize the units over the entire manuscript, especially for the oxygen. In Table 4, the sensor accuracy is given in µmol/kg (molinity) while in Section 3.2 the µmol/L (molarity) unit is used. The International System of Units in Oceanography (ISO) report published by Unesco (1985) recommended using moles per kilogram of solution for dissolved gases: "For concentrations of dissolved gases units such as milliliter or cubic centimeter per liter at the reference temperature and pressure should also be discarded. Such concentrations should henceforth be reported uniformly in moles per cubic decimetre, or in moles per kilogram of solution. (p.120)"*

Thanks for the suggestions, we use µmol/kg throughout the manuscript for O2.

*L. 315: "… 1053, respectively." Give the exact $R^2$ value.*

We have inserted a table (Table 2) concerning all the salinity statistics.

*L. 315: Give the exact $R^2$ value. What is the statistical test used, and between which variables? An $R^2$ of 0.6 is quite high, is it not? Sixty percent of the variance in oxygen can be explained by temperature variations, thus, if I understand well, I do not agree with the statement "independent of temperature". I also feel this sentence is written oddly (I don't get really what the temperatures refer to). I suggest splitting this sentence into two. Fig. 4e and f refer to oxygen concentrations, not oxygen saturations…*

This figure was removed in the new version of the manuscript. Surely a correlation of 0.6 is not low, I also agree that the text was not very clear Here we used a linear correlation between temperature and oxygen, what I wanted to highlight was the great variability observed at certain temperatures, such as 13.5°C. in order to justify that the oxygen variations were more due to a problem with the sensor than to natural variability

*L. 317: there is a space between °C and the comma*

The sentence is rewritten.

*L. 322: What is the criterion defining "unreasonable" data?*

The sentence is rewritten.

*L. 325: The authors stated (L. 322) that "no significant trend in sensors response" (according to Figures g & h it seems true) is observed (once again, what is the statistical test used here?) and then claimed that they want to "correct the negative trend". Please clarify and be consistent*

This is clarified now.

*L. 335: In the original vapor pressure of water equation (Johnson et al., 2015), the natural logarithm (ln) term is used while in the manuscript the logarithm (log) is written in the equation. Please check the equation and the calculations done.*

Sorry for it, we use the natural logarithm for computation, we corrected in the manuscript.

*L. 336: What is the unit for the volume fraction of oxygen, what is the value used? If it is a constant value (I believe that it might be 20.946 ± 0.002 percent), then the authors should write it*

We use the constant value, and we added it in the text.

*L.335, 338, 340 & 343: I suggest numbering the equations. Also, what are the units?*

We did that.

*L. 339: G is the gain factor/gain correction.*

Yes, we made it explicit in the text.

*L. 339: "The Epp was..."*

Thanks, we changed it.

*L. 341: The term SDcsd does not correspond to the what is written in the equation. I suggest clarifying this sentence here as follows "The corrected oxygen concentration (O2csd) from the SDs was calculated...". Data is a plural noun → "are/were" (and elsewhere in the manuscript). When SDs oxygen data not corrected are used, I suggest specifying "raw data" ("... from adjusting raw oxygen data measured by the SDs..").*

Thanks, we did that.

*L. 344: Here again, the sentence is very confusing and does not correspond to what was written previously. It is clearly stated in L. 316 that there is no temperature dependency... From my understanding, the purpose of the correction is not to detrend a time-series (that would imply a consistent drift/change over time, which is not the case here (Fig 4 g-h)) but to correct oxygen sensor instabilities and drifting over time as the gain factor is updated to correct for temporal drift.*

Thanks for the suggestions. We rewrite the sentence as follows:

For each transect the mean gain was calculated and then, the gain factor was multiplied by the hourly oxygen data allowing to correct the time series.

*L. 346: How these dissolved oxygen saturations deterministic (I assume) trends were removed?*

In this case the trend was removed by considering the ratio of air saturation to oxygen measurement. That is, for each measurement point of the SDs the gain factor was multiplied by the oxygen value, so all measurements were aligned with respect to air saturation. However, this limited the variability of the measurement, which is why

residuals were added (Fig. 4 g and h of the first version). This part has been removed in the new version of the manuscript.

*L. 348-353: Here again, it is unclear how dissolved oxygen data were detrended, how the relationships were calculated, what the statistical relevance of the calculations was... The last correction step (i.e. the biological activity effect removal) relying on the assumption that only biological processes drive the O2 content variability seems to induce an over-estimation of these processes as air-sea O2 fluxes and physical structures (convection, eddies) could significantly impact the O2 reservoir. A more detailed discussion (is it in agreement with the literature?) about this assumption (and its associated error) would be at least a way forward to add confidence. I am also unsure about the data used here... I would use the corrected oxygen data and then derive the residual values, as I have the feeling that otherwise an overestimation of the biological activity could occur.*

Thanks for all your suggestions. We decided to simplify the correction method for oxygen data similar to that of the Argo float correction.

*Section 3.2: The conversion between equilibrium partial pressure, pO2, and seawater O2 concentration, depends on the seawater salinity (see Bittig et al., 2016 & 2018). Was this salinity correction done?*

The salinity correction was performed as for temperature and pressure, before the O2 correction.

*L. 360: Remove "the" Spring 2020.*

This sentence is rewritten.

*L. 361: Why is it "particularly interesting"?*

The sentence is rewritten.

*L. 366: I would remove the term "sensor" before "pCO2 measurements" as it could lead to confusion with the SD pCO2 system and suggest another term like "fixed-sites pCO2 measurements" (feel free to ignore this comment, this is a matter of personal perception). At least please harmonize the nomenclature over the entire paragraph (line 380 they are called "stations sensors")*

Thanks, the nomenclature is harmonized.

*L. 368: "... Atlan2Med cruise/campaign/experiment"*

The sentence is modified.

*L. 370: Please develop this "minor variability"*

We have referred to Table S2 and S3¤, which explains the different frequencies.

*L. 376: Depending on the input variables/combination used to derive the pCO2 (as well as on the equilibrium constants), the uncertainty could vary a lot, from ± 1.8 µatm to ± ~6 µatm (e.g., Millero,*

*1995, Orr et al., 2018). Please justify your choices.*

We have added some more info regarding this.

*L.377: Please develop the "the hydrogen fluoride constant KF" and put the F as a subscript*

This is done.

*L. 378: Please put the letter "p" in italics (and elsewhere in the manuscript)*

Done.

*L. 394: "... (in µatm)"*

Done.

*L. 397: Considering Figure 6b, is it still accurate to consider an associate uncertainty of ± 5 µatm?*

The uncertainty of 5 uatm is estimated after in-lab corrections. However, we are aware that the drift over the demonstration experiment amounted to more than 10 uatm.

*Section 3.3 and 3.4: I would suggest putting them as sub-sections (3.3.1 & 3.3.2) as a bigger one merging them and entitled, for example, "Correction and adjustment of pCO2 data". I would thus rename Section 3.3.1. to be more precise "Fixed-sites pCO2 data acquisition and qualification".*

Thanks for the suggestions, we modified the paragraphs as recommended.

*Results and discussion*

*Section 4.1: For salinity data, the correction is discussed in this "results and discussions" section while for the other parameters (oxygen and pCO2), corrected data are already presented and discussed in the Methods section. Please harmonize the data presentation.*

The manuscript has been restructured to meet this request.

*L. 404: Add a comma after "(T1)"*

The sentence has been rewritten.

*L. 407: "The salinity correction induced an over-estimation /over-estimated the salinity values. Instead, raw salinity data were in ..."*

The sentence has been rewritten.

*L. 411: Please add a reference to the plot you are referring to.*

Done.

*L. 411: "with respect to"*

Done.

*L. 414: "... (Fig. 7e). The corrected salinity ..."*

The sentence has been rewritten.

*L. 418: "... between the E2M3A fixed ocean station ...". In this sentence, you are comparing a structure itself to data. Please be clear: "fixed ocean station dataset and glider measurements"*

We have clarified this.

*L. 420: "poor": I suggest removing those kinds of subjective adjectives that are uncountable and sound not scientific. Replace by "non-significant" and present a strong argument (statistics)*

Following your suggestions, we removed all of this subjective adjectives in the manuscript

*L. 422: This plot does not show trends! Please use the words temporal changes, variations, (similar) dynamics*

Thanks, we considered your suggestions in rebuilding the text.

*L. 423: "... June 1$^{st}$ ... April 27$^{th}$ ..."*

In this manuscript, we have chosen to use the date format 1 June 2020 all over.

*L. 426: "... are consistent... ; differences being mainly due to the distance ..."*

The sentence has been rewritten.

*L. 428: "Considering that during T1 ...to apply the salinity correction after this transect."*

The sentence has been rewritten.

*L. 432: "... discrete dissolved oxygen measurements were not available ... Thus, the SD corrected ..."*

The sentence has been rewritten.

*L. 435: Please name the two locations chosen*

We specify it in the text.

*L. 436: L. 235, the authors claimed that Chl-a values showed inconsistencies, while here they are used. It leads to confusion: I suggest moving L.460 upward. Moreover, this analysis comes off as very simplistic as it relies only on a visual inspection.*

This is modified as suggested.

*L. 439: Please add a reference to support this statement*

Done.

*L. 440: Here again, please add concrete data to support this "strong linear correlation" statement*

The sentence is modified.

*L. 441: What does this "slightly lower/higher" represent? Please add a reference*

The sentence is modified.

*L. 444: Is it always the case? Even during intense wind episodes? (Ulses et al., 2020)*

The sentence is modified, and reference included.

*L. 446: Are the authors describing here the oxygen saturation in the Mediterranean Sea? I suggest applying this example/sentence to Mediterranean Sea data*

The part is modified.

*L. 449: How is the evaluation done?*

This part is modified and more explanation is added.

*L. 451-452: The DOI can be removed from the main text and put in the data availability section*

We did it

*L. 456: Same remark here for the dates than at Line 423*

In this manuscript, we have chosen a different date format.

*L. 457: were high*

This is done.

*L. 458: "... lines). The optical sensors on the SDs..." L. 461: "... needs weeks .."*

This part is modified.

*L. 462. "We refer to...". Please remove the thus*

This part is modified.

*L. 464: Please explain the link between Chl-a data and the temperature, or at least modify the sentence as you are starting with "the patch of high Chl-a" and continuing with "evident in sea surface temperature".*

Done.

*L. 482: The decrease in oxygen is not observed for both SDs, only on the SD 1053 time-series.*

Thanks for the comment. Checking further, I realized that on 1 April, the sensor cleaning was done on the vehicle, so the rapid decrease could be due to this operation. In the new figure I did not include data from 1 April. What I wanted to show was the oxygen undersaturation in that area due to the cyclonic circulation.

*L. 488: A reference would be appreciated here to support this statement*

Text is modified and reference inserted.

*L. 494: A brief discussion about this relationship, varying regionally, and its consequences (i.e., the T sensitivity is larger in colder regions and lower in the warmer tropics) would maybe be interesting here (see Gallego et al., 2018)*

Thanks for this suggestion, however, we consider this being out of the scope of this paper. We have included a reference to the relationship.

*L. 500: What were the input variables at the other sites? What would be the uncertainty associated with the other difference estimates?*

The input variables are listed in Table S4.

*L. 504: Would it be possible to estimate the impact of those processes?*

This is difficult, which we have stated in the text

*Data availability*

*Section 5 & Table 8: I suggest moving this section and the table to the Supplementary Material. In Table 8, please split the DOI column into two columns, one only with the DOI and the other with the references.*

Done.

*Summary*

*L. 519: "... covering both the Eastern Tropical North Atlantic region ...". Also, as the ETNA region is cited numerous times, maybe the acronym could be used here and previously*

Thanks for this suggestion, we have used the acronym in the manuscript

*L. 529: sensors on the SDs (also L. 531, 556, 562). Moreover, what is corrected is not the sensor itself but the data produced by it. Please correct*

Thanks for the suggestion, we considered it in the revised version of the manuscript.

*L. 531: I do not agree with this sentence: the oxygen data correction method is not new (Johnson et al., 2015, Bittig et al., 2018)*

We agree with your suggestion, and we removed the sentence.

*L. 533: "... data sets that are available...". Please harmonise the way the word dataset is written over the entire manuscript (sometimes with a space, sometimes no)*

Thanks for the suggestion, we have harmonized the word throughout the manuscript (dataset).

*L. 535: the paragraph begins with a reading out of the limitations, but it seems that there is only one issue considering the rest of the paragraph. I suggest rephrasing or modifying the first word*

The sentence is rewritten.

*L. 536: "... cruises, etc."*

This is rewritten.

*L. 538: "Consistent" is an overstatement as no statistics are associated with the results. I also suggest rephrasing the sentence as it is more or less written two times that the consistency is fine…*

Thanks for the suggestions, we considered this in the new version of the manuscript.

*Future and recommendations*

*L. 549: I would move the beginning of this Section from line 544 to 549 in the conclusion section, reducing its content and rephrasing some parts of the conclusion to fit in. Indeed, no recommendations or pieces of advice are given here, only a repetition of the previous statements*

This is reorganized.

*L. 556: The RBR acronym is not defined in the manuscript*

We have added a link to the company.

*L. 565: Please rephrase... "the sensor mounting should ensure that the sensors are mounted"*

*sounds repetitive*

This is changed.

*L. 566: "... from a lack of ..."*

This is changed.

*For this section, I would suggest summarising the recommendations via a bullet list or at least by numbering them. It would clarify what the improvements from this experiment are based on your observations, which is unclear in the current Section 7*

Thanks, following your suggestions, we modified the paragraphs.

*** Figures & Tables ***

Thanks for the suggestions, all the figures were modified in the new version of the manuscript

*Figure 1: The bottom panel seems to have been warped. Please add on this one some indications about the oceans/seas, countries, and the lat/long information for the x/y axis. For the upper panel, the legend is incomplete: Argo float symbols and the two SDs colors are detailed, but the symbols for the fixed-stations, the glider, and the research vessel are not explained. I suggest adding on the small maps the names of the fixed sites and small dashed lines to link the "zoom windows" to the small squares on the map. I also suggest shifting them a little bit to better show the main map.*

*The two blue lines (T1 & T2) are superimposed, but I thought the two SDs were deployed at the same time… Also, the transect cutting is incomplete: to what are the data (SD 1030) between December and January qualified/associated with?*

*Table 2: Not needed, could be completely removed and replaced by 2 sentences in the main manuscript. Also, the backslash (\) is not in the same direction as the others in the manuscript (column 1 first box)*

*Table 3: In the main text, SBE16 plus is written with a +. Please standardize. Make uniform the units too, with either a -1 as a superscript or a /*

*Figure 2: Would be useful to add on the Figure the transects (T1, T2, …). Please modify the legend ".. and Argo float data..". The legend on the figure (i.e. stations) is confusing as it could refer to the fixed station sites described before. Please clarify. I also suggest adding a sentence in the main manuscript stating that in the manuscript salinity values are expressed in PSU (then remove this unit in the figures).*

*Figure 3: The y-axis is not labeled (ΔS?). Also, I suggest harmonizing the y-axis range/scale to avoid confusion. What is the statistical test used? See my previous comment about that. Colors for each SD are not the same between Figure 2 and Figure 3 (blue and purple against black and red): please harmonize. A map on the side would be relevant to better visualize where the data were acquired.*

*Figure 4: It seems that the red color is used for the SD 1053 and the black one for the SD1030, but in panels g and h it is the reverse. Please harmonize. I suggest reducing this Figure by (1) putting the temperature panel with Figure 2 (as temperature data are cited earlier in the paper at the beginning of Section 3, (2) removing the panels e and f (not necessary, could be put in the Supplementary Material) and (3) moving upward the panels e and f as this section is first and foremost dedicated to oxygen data. I also suggest adding the statistical results to these panels. In Figure 4b, please reduce the thickness of the black trend line as the grey one is currently not visible. The y axes are not completely labeled (Oxygen Saturation (%), O2 (μmol/l)). What are the dashed grey lines?*

*Figure 5: The y axes are not completely labeled (Oxygen Saturation (%), O2 (μmol/l), temperature...). In Figure D, the legend box is on the x-line. I suggest reducing a lot this figure by removing some panels (e.g. the detrended and not detrended, especially as it is unclear in the main manuscript...). Once again here, I suggest moving the "corrected O2 data" time-series plot to the top as this is what this section is about. Please harmonize the legend in the boxes (correction/corrected...). In the legend, please correct "back → black"*

*Figure 6: I suggest completing the y-axis for panel b by adding Δ (i.e. ΔpCO2 (μatm)) to avoid any confusion with panel A.*

*Figure7: Colors are almost indistinguishable on the upper panel (Fig. 7a). I also do not think that it is relevant here to plot the model salinity data as SD data were corrected using them... They obviously more or less match them. I suggest not presenting the data with multiple panels as it is done now (corresponding to certain transects) but rather using the current panels A and B and adding the other reference measurements on them. If they are bigger the dots will be big enough and the main message clear. Thus, it would delete the panels c to g. Please also use another for the floats and the glider data (both of them are yellow…). Following the text structure, I would first present the data recorded by the SD 1053 (first discussed) and then by the SD 1030. Numbers on the panels (corresponding to the distances) are unreadable. If the authors want to keep the panels as they are currently, I suggest at least using the same y-axis scale (Fig. 7 c-e).*

*Figure 8: Units are missing. The y-axes as well as the colorbar are incomplete. The legend is incomplete too as the second y-axis in Figure 8b is not explained.*

*Figure 9: One panel is enough for both the sea surface temperature and the vertical section. It is not needed to put the three dates as the plots are similar. Here too, units are missing. The y-axes as well as the colorbar are incomplete. The addition of the SDs pathways on panel C would be useful to understand the differences in O2 concentration observed between the two platforms.*

*Figure 10: Are fCO2 data used or pCO2? The legend is not in agreement with the plots and Section 4.3. In the boxes, please write pCO2 rather than CO2. In Figure 10B, why dots (that are indistinguishable) are not linked?*

*Table 8: A "E" is missing line 6 column 3 "MOOS program". I suggest renaming the first column "Platform" or similar*

*Generally, the number of plots (that is too high!) for each parameter studied is not well balanced. Even if it does not have to be equal, there is a strong disproportion among the figures, with for example 4 figures for the salinity, 15 for the oxygen, and only 2 for the pCO2 in the methods section.*

We have improved the figures as suggested by the referee and ensured that there is a balance between the salinity, dissolved oxygen and pCO2 figures in the manuscript.

*General remark: Please have a look at the colors used for all figures and make sure they are colorblind-friendly (I do believe that it is not the case for Figures 8 & 9).*

We change the colormap using the cmocean tool.

---

## Referee Report (RR1)

**Review of *CO₂ and hydrography acquired by Autonomous Surface Vehicles from the Atlantic Ocean to the Mediterranean Sea: data correction and validation* by Martellucci et al. (essd-2023-457)**

**Overview**

Martellucci and coauthors describe data collected by two Saildrone (SD) autonomous surface vehicles (ASVs) in the Atlantic Ocean and Mediterranean Sea during the ATL2MED experiment in 2019 and 2020. In particular, they discuss strategies to correct salinity, dissolved oxygen, and $pCO_2$ data from the ASVs, and compare their corrected datasets to a variety of validation datasets from cruises, fixed moorings, Argo floats, and gliders.

This manuscript represents an important contribution to the field, as measurements from autonomous systems are becoming a critical way to fill observational gaps in hydrographic and biogeochemical ocean observations. The demonstration of creative and effective methods to ensure data delivered by these autonomous systems are scientifically credible is therefore an essential step in filling those gaps. The importance of this work is particularly underscored by its context within the timeline of the COVID-19 pandemic. The difficulty conducting regular maintenance and collecting validation measurements experienced during the ATL2MED experiment due to pandemic restrictions might serve as a good analogue for the future of autonomous observations, when many more autonomous platforms may be deployed throughout the global ocean and cost considerations could limit comprehensive maintenance plans and the number of independent measurements available for comparison.

The manuscript is generally well written and contains sufficient information describing data collection, adjustment, and analysis. I'll detail a few minor comments below that could help make this a stronger contribution.

**General comments**

In the discussion section, analysis of the corrected salinity and dissolved oxygen datasets in their oceanographic contexts is provided, but this analysis is conspicuously absent from the $pCO_2$ section. Instead only apparent differences between SD $pCO_2$ and $pCO_2$ from fixed stations is discussed. I'd suggest augmenting this section with some brief analysis; for example, what causes the increase in sea surface $pCO_2$ toward the end of the experiment?

The 'Experiences and recommendations' section feels rather hastily written to me. The suggestions provided are good ones, but are sometimes repetitive and delivered in a way that is somewhat difficult to follow. I'd suggest revising this section for clarity.

**Line-by-line comments**

52-53: Remove 'the' before biofouling: '…one of the most important is biofouling…'

Figure 1 caption: should this be 'glider sections'? Also, section is spelled incorrectly within the figure

102: Perhaps also cite Sabine et al. (https://doi.org/10.1175/JTECH-D-20-0010.1) when discussing the ASVCO2 system

111, 121: Not sure what is meant by 'open fixed station'. Should this be 'open-ocean'?

127: Define OGS here (this is the first mention in the text, besides the author affiliations). Also, in the next line, the authors have 'the OGS'. I'd recommend consistency: either 'the OGS' or just 'OGS'.

201-202: Grammar. Change 'allowing to' to 'allowing for the correction of'

275: Grammar. Change 'less' to 'fewer'

300: Can it be specified what level of uncertainty would have been assumed in the absence of these issues with the ASVCO2 instrument?

Figure 8d: x-axis label should be 'Latitude'

Figure 9b: y-axis label should be '$\Delta p\text{CO}_2$'

Figure 10b: Can a line be added through the W1M3A values, like for the other datasets?

430-439: No mention of the $p\text{CO}_2$ results in this paragraph?

436: Could this be better stated as 'consistency in the corrected salinity values between both SDs'?

437: Could this be better stated as '...to correct the erroneous trend in $\text{O}_2$ saturation %'?

455: Typo. 'substantial' amount of effort

474: This is the first mention of RBR. Are these the optical sensors?

497: Author contribution section is incomplete

---

## Author Response (AR2)

We thank the referee for the careful and insightful review of our manuscript and for all your suggestions and recommendations. Your work will help to improve the paper significantly. We took into account all the major points and concerns addressed by the reviewer. In the revised manuscript, the edited text is highlighted in red. We hope that this revised version is in agreement with your suggestions and suitable for publication in Earth System Science Data.

**General comments**

*In the discussion section, analysis of the corrected salinity and dissolved oxygen datasets in their oceanographic contexts is provided, but this analysis is conspicuously absent from the pCO2 section. Instead only apparent differences between SD pCO2 and pCO2 from fixed stations is discussed. I'd suggest augmenting this section with some brief analysis; for example, what causes the increase in sea surface pCO2 toward the end of the experiment?*

Thanks for the comment. A new paragraph and figure were inserted into the text to explain the Co2 trend throughout the experiment. To better understand the variability of $pCO_2$ this was compared with Temperature and Chlorophyll a data. In addition, the analysis of the processes observed throughout the experiment is the subject of a work in progress that will discuss $pCO_2$ variability in depth. Below the added paragraph.

To assess the representativeness of the pCO2 correction in terms of ecosystem dynamics, a comparison was made between the corrected pCO2, temperature, and Chl-a concentrations from satellites. The pCO2 in seawater is influenced by primary production, respiration, air-sea gas exchange, formation and dissolution of calcium carbonates, water mixing, riverine discharges and advection (Zeebe and Wolf-Gladrow, 2007; Bauer et al., 2013; Millero 2007), which leads to significant variations in different regions. The temperature affects the pCO2 through the thermodynamic dissociation constants of the carbonic acids, which directly affects the CO2 equilibria (eg. Millero, 2007) and to a lesser extent also the gas solubility.

[revised manuscript text omitted]

**Line-by-line comments**

*52-53: Remove 'the' before biofouling: '…one of the most important is biofouling…'*

Thanks for the suggestion, we removed the in the sentence.

*Figure 1 caption: should this be 'glider sections'? Also, section is spelled incorrectly within the figure*

Thanks we modified the figure accordingly with your comment.

*102: Perhaps also cite Sabine et al. ( ht tps: //doi.org/1 0 . 1 17 5 /JTECH -D -20 -0 0 10.1 ) when discussing the ASVCO2 system*

Thanks for the correction. The references were added in the text.

*111, 121: Not sure what is meant by 'open fixed station'. Should this be 'open-ocean'?*

Thanks, we change open fixed station in open -ocean.

*127: Define OGS here (this is the first mention in the text, besides the author affiliations). Also, in the next line, the authors have 'the OGS'. I'd recommend consistency: either 'the OGS' or just 'OGS'.*

I apologise for not define OGS. In the main text we define it, and in the manuscript we use only OGS.

*201-202: Grammar. Change 'allowing to' to 'allowing for the correction of'*

Thanks for the suggestion, we modified the sentence accordingly to your comment.

*275: Grammar. Change 'less' to 'fewer'*

Thanks for the correction, we change less with fewer.

*300: Can it be specified what level of uncertainty would have been assumed in the absence of these issues with the ASVCO2 instrument?*

Thanks for the suggestion, we included a new sentence specifying the uncertainty. Below the added sentence.

Laboratory tests of the ASVCO2 system on SD platforms highlighted an uncertainty of less than 2 µatm (Table 3 in Sutton et al., 2014).

*Figure 8d: x-axis label should be 'Latitude'*

I apologise for this, we change Longitude with Latitude.

*Figure 9b: y-axis label should be 'ΔpCO2'*

Thanks for the correction. The label were changed.

*Figure 10b: Can a line be added through the W1M3A values, like for the other datasets?*

Thanks for the suggestion we modified the figure accordingly to your comment.

*430-439: No mention of the pCO2 results in this paragraph?*

We included a new paragraph (see the first general comment).

*436: Could this be better stated as 'consistency in the corrected salinity values between both SDs'?*

Thanks we modified the figure accordingly with your comment.

*437: Could this be better stated as '...to correct the erroneous trend in O2 saturation %'?*

Thanks we modified the figure accordingly with your comment.

*455: Typo. 'substantial' amount of effort*

Thanks for the suggestion, we add substantial on the sentence.

*474: This is the first mention of RBR. Are these the optical sensors?*

RBR is a company that manufactures oceanographic sensors, from physical to biogeochemical. For this experiment, RBR supplied sensors to measure temperature, salinity, dissolved oxygen and chlorophyll a. Oxygen and chlorophyll a are optical sensors. These sensors did not work properly during the entire mission, so it was decided to exclude them.

*497: Author contribution section is incomplete*

I apologise for not including it. The paragraph was included in the text.